# Dopamine drives *Drosophila sechellia* adaptation to its toxic host

**Sofía Lavista-Llanos[1]\*, Aleš Svatoš[1], Marco Kai[1†], Thomas Riemensperger[2‡], Serge Birman[2], Marcus C Stensmyr[1§], Bill S Hansson[1]**

[1]Max Planck Institute for Chemical Ecology, Jena, Germany; [2]Genetics and Physiopathology of Neurotransmission, Neurobiology Unit, CNRS, ESPCI ParisTech, Paris, France

**Abstract** Many insect species are host-obligate specialists. The evolutionary mechanism driving the adaptation of a species to a toxic host is, however, intriguing. We analyzed the tight association of *Drosophila sechellia* to its sole host, the fruit of *Morinda citrifolia*, which is toxic to other members of the *melanogaster* species group. Molecular polymorphisms in the dopamine regulatory protein Catsup cause infertility in *D. sechellia* due to maternal arrest of oogenesis. In its natural host, the fruit compensates for the impaired maternal dopamine metabolism with the precursor l-DOPA, resuming oogenesis and stimulating egg production. l-DOPA present in morinda additionally increases the size of *D. sechellia* eggs, what in turn enhances early fitness. We argue that the need of l-DOPA for successful reproduction has driven *D. sechellia* to become an *M. citrifolia* obligate specialist. This study illustrates how an insect's dopaminergic system can sustain ecological adaptations by modulating ontogenesis and development.

\*For correspondence: slavista-llanos@ice.mpg.de

Present address: †Institute for Biological Science, Department of Biochemistry, University of Rostock, Rostock, Germany; ‡Molecular Neurobiology of Behaviour, Johann Friedrich Blumenbach Institute, Georg August University of Goettingen, Goettingen, Germany; §Department of Biology, Lund University, Lund, Sweden

## Introduction

*Morinda citrifolia* fruit (morinda) is the sole host of *Drosophila sechellia* (*Tsacas and Baechli, 1981*), a close relative of *Drosophila melanogaster* and endemic to the Seychelles archipelago (*Louis and David, 1986*). A peculiar aspect of the specialization is that morinda fruits are toxic to all other drosophilids (*Legal et al., 1992*). The toxicity stems from a high content of carboxylic acids (primarily octanoic and hexanoic acid) (*Legal et al., 1994*), to which *D. sechellia* appears to be resistant (*Farine et al., 1996*). The chemosensory system of *D. sechellia* is specialized in detecting and coding key volatiles produced by morinda (*Dekker et al., 2006*) while devoid of the repellence towards the acids (*Matsuo et al., 2007*). On the other hand, *D. sechellia* females exhibit a low reproductive potential, partly because of a low ovariole number and partly because of fairly low egg production (*R'Kha et al., 1991*; *R'kha et al., 1997*), making it difficult to raise *D. sechellia* under laboratory conditions. In turn, morinda stimulates egg production (*R'kha et al., 1997*), and *D. sechellia* clearly prefers to oviposit in medium containing morinda carboxylic acids (*Amlou et al., 1998*). On its host, *D. sechellia* increases expression of genes involved with oogenesis and fatty acid metabolism (*Dworkin and Jones, 2009*). Thus, we here examined the dependence of *Drosophila sechellia* on morinda, for optimal reproduction.

All animal embryos rely on maternally provided gene products for their initial development prior to zygotic genetic synthesis. Maternal effects can thus act as a form of cross-generational phenotypic plasticity, playing a role in an animal's adaptation to toxic environments. Embryonic survival in morinda is a maternally inherited trait and does not depend on the genotype of the embryo (*R'Kha et al., 1991*). We therefore considered if maternal effects sustained the evolutionary process that has driven the specialization of *D. sechellia*. Results enhance our understanding of the reproductive behaviour of *Drosophila* and suggest an ontogenetic mechanism of insect adaptation to a toxic host.

**eLife digest** Many insect species rely on another animal or plant species for their own reproduction. For example, a fruit fly called *Drosophila sechellia*—which is found in the Seychelles—will only feed and lay its eggs on the fruit of a species of tree called *Morinda citrifolia*. This pairing is particularly unusual because these fruits, commonly called morinda, are toxic to all other *Drosophila* species.

Female *Drosophila sechellia* flies produce fewer eggs than other *Drosophila* species, which makes it difficult to raise this species in the laboratory. However providing these flies with morinda fruit, or chemicals from this fruit, was known to increase the expression of many genes involved in egg production and stimulate the flies to lay more eggs. Nevertheless, the reasons why this species of fruit fly depends on the toxic morinda fruit were unclear.

Now Lavista-Llanos et al. have confirmed that feeding *Drosophila sechellia* flies a diet of morinda fruit—instead of a typical laboratory diet—causes these flies to produce six-times as many eggs. Furthermore, this morinda diet had effects that went beyond the previously reported stimulatory effects of acidic chemicals in the fruits triggering the flies to lay more eggs.

Egg production in flies is controlled by dopamine, and a lack of this hormone is known to reduce the size of other fruit flies' ovaries and the number of eggs that they produce. Lavista-Llanos et al. went on to feed female *Drosophila sechellia* flies the chemical building blocks that make up the dopamine hormone, and one such chemical (called l-DOPA) caused the flies to produce more eggs. This did not occur when the flies were fed dopamine itself.

Lavista-Llanos et al. discovered that *Drosophila sechellia* flies have very high levels of dopamine but much lower levels of l-DOPA than other *Drosophila* fly species; and revealed that this was because a gene called *Catsup* is mutated in *Drosophila sechellia*. When Lavista-Llanos et al. mutated the same gene in another *Drosophila* species, the mutant flies produced fewer eggs and abnormally accumulated an enzyme (which makes l-DOPA) inside their developing eggs—just like *Drosophila sechellia*.

The presence of l-DOPA in morinda fruit partly compensates for the reduced fertility of *Drosophila sechellia* and the other flies with mutations in the *Catsup* gene. Lavista-Llanos et al. discovered that removing or replacing l-DOPA in the morinda fruit caused the flies to produce fewer eggs. Furthermore, the l-DOPA present in morinda increases the size of *Drosophila sechellia* eggs, which in turn helps them to survive their toxic environment.

Lavista-Llanos et al. also discovered that feeding dopamine to vulnerable *Drosophila* species helps them to cope with the toxic effects of a morinda diet. One of the next challenges will be to uncover how chemicals from the morinda fruit affect the dopamine system of the flies. It is also unknown if the dopamine hormone also influences the strong attraction that *Drosophila sechellia* feels towards its only host, the morinda fruit.

## Results and discussion

We first examined the influence of chemicals found in morinda on the reproductive system of *D. sechellia* by testing the effect of diet on egg production. We raised two geographically different populations of *D. sechellia* on standard *Drosophila* cornmeal medium (standard diet), on morinda fruit (morinda diet), or on a non-host fruit (banana diet). For comparison, we also raised two strains of *D. melanogaster* (wild-type Berlin and Canton-S) on the same media (see 'Materials and methods'). To score the rate of egg production, we transferred the flies to oviposition cages containing agar plates devoid of oviposition stimuli (i.e., yeast or morinda carboxylic acids). In agreement with previous reports, *D. sechellia* raised on standard diet produced few eggs compared to *D. melanogaster* wild-type Berlin and *D. melanogaster* Canton-S (*Figure 1A* and *Figure 1—figure supplement 1*), raised on the same media. The addition of morinda carboxylic acids to the standard diet did not increase egg production in *D. sechellia* (*Figure 1B*). *D. sechellia* raised on a non-host fruit (banana diet) showed a slight increase in the number of eggs laid (*Figure 1B*), but this number did not increase further with the addition of morinda carboxylic acids (*Figure 1B*). *D. sechellia* raised on morinda diet, however, showed a sixfold increase in egg production (*Figure 1B* and *Figure 1—figure supplement 1*). Morinda diet did not affect the number of eggs produced by *D. melanogaster* wild-type Berlin and

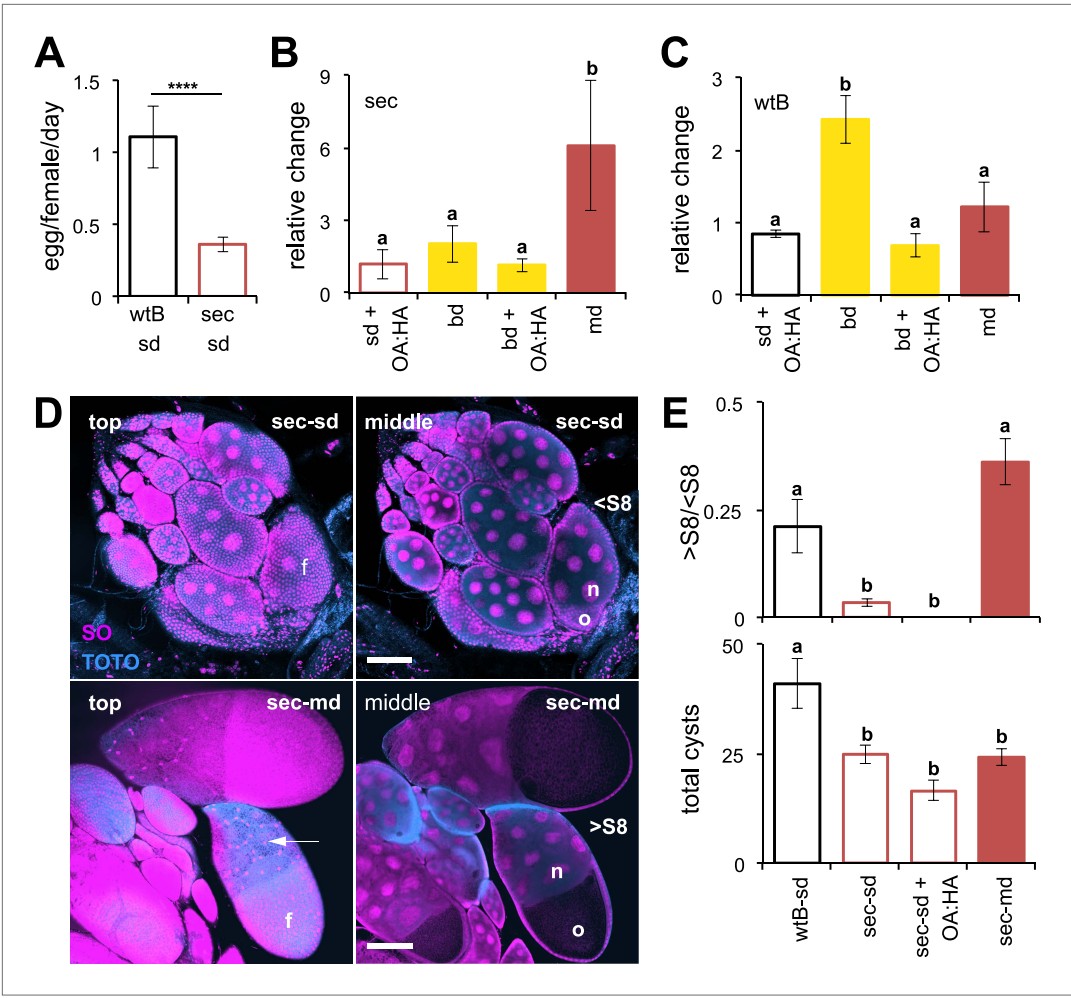

**Figure 1**. Morinda increases egg production in *D. sechellia*. (**A**–**C**) Egg production (egg/female/day) (*N* > 20) (**A**) and its relative change (*N* > 5) (**B** and **C**) in *D. sechellia* (14021–0248.25, sec) and *D. melanogaster* wild-type Berlin (wtB) fed a standard diet (sd), or morinda diet (md), or banana diet (bd), or diets supplemented with morinda carboxylic acids (+OA:HA). (**D**) Confocal images showing the surface (top) or the interior (middle) of ovarioles stained with nucleic acid specific dyes (sytox orange (SO) and TOTO) of *D. sechellia* (14021–0248.25) fed a standard diet (sec-sd) or a morinda diet (sec-md). The follicle cells (f) surrounding the oocyte (o) or stretched over the nurse cells (n) (arrow) are indicated for early (<S8) or vitellogenic cysts (>S8). Scale bar 100 μm. (**E**) Rate of vitellogenesis (>S8/<S8, top graph) and number of cysts (total cysts, lower graph) (*N* > 8) in *D. sechellia* (14021–0248.25, sec) and *D. melanogaster* wild-type Berlin (wtB) fed a standard diet (sd), or a morinda diet (md), or a diet supplemented with morinda carboxylic acids (+OA:HA). Different letters denote significant differences (p < 0.01) using ANOVA followed by Tukey's test (**B**–**E**); ****p < 0.00001 using Student's *t* test to compare species (**A**). Error bars represent s.e.m.

The following figure supplements are available for figure 1:

**Figure supplement 1**. Morinda stimulates egg production in *D. sechellia*.

**Figure supplement 2**. Apoptosis in *D. sechellia* ovaries.

**Figure supplement 3**. Feeding behavior in *D. sechellia*.

slightly reduced those of *D. melanogaster* Canton-S (*Figure 1C* and *Figure 1—figure supplement 1*), as did the addition of morinda carboxylic acids to the standard diet (*Figure 1C*). Banana, however, tripled egg production in *D. melanogaster* wild-type Berlin (*Figure 1C*). This increase could be inhibited by the addition of morinda carboxylic acids to the banana (*Figure 1C*). These results demonstrate

that the presence of a natural host modulates the reproductive capacity of *Drosophila*. In particular, morinda has a strong effect on *D. sechellia* egg production that goes beyond the reported stimulatory effect of the carboxylic acids on oviposition (*Amlou et al., 1998*).

To study the mechanism behind the dietary modulation of egg production, we next examined oocyte development. The germarium in each ovariole continuously produces oocyte-cysts, each composed of 15 nurse cells and one oocyte, surrounded by a layer of follicular cells (*Figure 1D*). Each oocyte-cyst follows 14 stereotypical consecutive stages of pre-vitellogenic (S1–S7) and vitellogenic (S8–S14) growth, each easily distinguishable by morphological criteria (*Spradling et al., 1997*). Mated *D. melanogaster* females typically exhibit multiple ovarioles carrying vitellogenic cysts (*Spradling et al., 1997*). Females of *D. sechellia* had cysts normally composed of 15 nurse cells and one oocyte, surrounded by follicular cells (*Figure 1D*). Kept on standard diet, mated *D. sechellia* held oocytes in early developmental stages (<S8) (*Figure 1D*) with few ovarioles, if any, carrying vitellogenic cyst (>S8), what resulted in a significantly lowered rate of vitellogenesis (calculated as >S8/<S8) in *D. sechellia* compared to *D. melanogaster* wild-type Berlin (*Figure 1E*). The low number of ovarioles in *D. sechellia* (*R'kha et al., 1997*) was reflected in the low number of total cysts compared to in *D. melanogaster* wild-type Berlin (*Figure 1E*). The halt in oocyte development explains the low production of eggs in *D. sechellia* raised on standard diet. *D. sechellia* fed on morinda diet showed a significantly increased vitellogenesis rate (*Figure 1D,E*). Adding the morinda carboxylic acids, however, had no effect on vitellogenesis in *D. sechellia* (*Figure 1E*). We thus conclude that *D. sechellia* requires components of the morinda fruit other than just the carboxylic acids to induce vitellogenesis and increase egg production.

Environmental and physiological stressors can cause egg chambers in the ovaries of *Drosophila* to be eliminated by apoptosis at oogenesis checkpoints in region-2/3 of the germarium or cyst stage S7/S8 (the mid-oogenesis checkpoint) (*Drummond-Barbosa and Spradling, 2001*; *McCall, 2004*). To analyze if apoptosis occurs in *D. sechellia*, we stained live ovaries with the vital dye acridine orange (*Arama and Steller, 2006*). Flies fed a standard diet showed massive apoptosis occurring at S7/S8 cysts, and to a lesser extent in region-2/3 of the germarium (*Figure 1—figure supplement 2*); suggesting that apoptosis could be the reason for the depressed oviposition on standard media. On the other hand, *D. sechellia* flies fed a morinda diet showed strikingly few apoptotic cysts (*Figure 1—figure supplement 2*), in line with the stimulatory effect of morinda in egg-production. Low food availability can trigger the mid-oogenesis checkpoint (*Terashima and Bownes, 2004*). Thus, we asked if the observed apoptosis in *D. sechellia* raised on standard diet was a consequence of starvation or aberrant feeding behaviour. In fact, *D. sechellia* adults raised on the standard diet showed slightly higher weights (*Figure 1—figure supplement 3*), and also consumed more food than did flies raised on morinda diet (*Figure 1—figure supplement 3*). Hence, we conclude that the halt in oogenesis observed in *D. sechellia* is not a consequence of starvation or aberrant feeding behaviour.

Oocyte-cyst progression is under tight maternal hormonal regulation: juvenile- and steroid hormones are mutually controlled by the biogenic amine dopamine (DA), ensuring normal oogenesis (*Gruntenko and Rauschenbach, 2008*). Interruption of DA during development in *D. melanogaster* results in small ovaries and poor egg production (*Neckameyer, 1996*), both of which are features that characterise adult *D. sechellia* (*R'kha et al., 1997*). Moreover, genes involved in DA differentiation were shown differentially expressed in *D. sechellia* compared to the generalists species *D. melanogaster* and the sister *Drosophila simulans* (*Dworkin and Jones, 2009*; *Wurmser et al., 2011*). Accordingly, we wondered if *D. sechellia* suffers from a DA deficiency. Therefore, we first tested if the addition of monoamines was sufficient to rescue egg production by feeding *D. sechellia* standard diet supplemented with amines and amine precursors. Indeed, supplementing the standard diet with the DA precursor 3,4-dihydroxyphenylalanine (L-DOPA) significantly increased the number of eggs produced by *D. sechellia* (*Figure 2A*), whereas the addition of DA itself had no effect (*Figure 2A*). Likewise, adding the monoamine octopamine (OA), or its precursor tyramine (TA), did not affect egg production (*Figure 2A*). L-DOPA supplementation did not change the total number of *D. sechellia* oocyte-cysts (p = 0.28103, one-way ANOVA). Instead, it drastically reduced apoptosis in the ovary (*Figure 2B*) increasing vitellogenesis significantly (*Figure 2C*), whereas DA, TA and OA had no such effect. These results show that the administration of the DA precursor L-DOPA is sufficient to stimulate oocyte progression and egg production in *D. sechellia*.

The findings so far thus led us to speculate that morinda contains monoamines, which stimulate oogenesis in *D. sechellia*. Indeed, we detected 180.4 ± 3.5 ng L-DOPA (mean ± s.e.m; N = 3) per gram

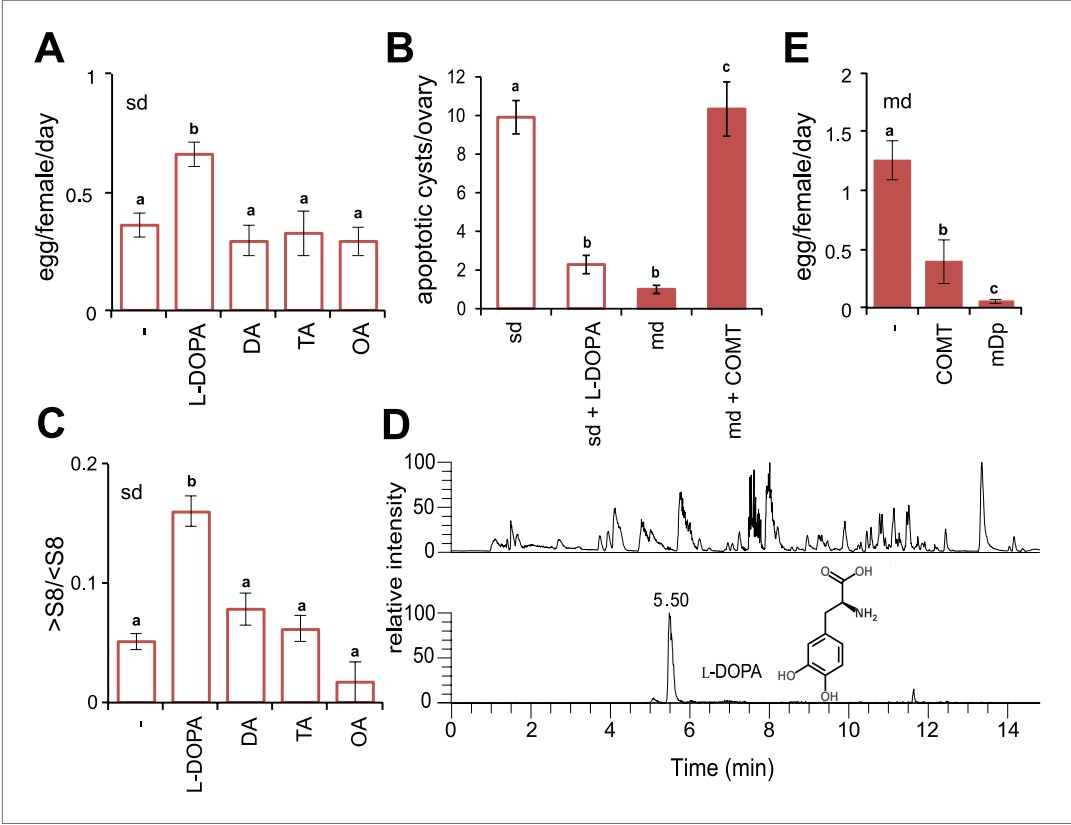

**Figure 2**. Morinda L-DOPA is required to stimulate egg production. (**A**) Egg production (egg/female/day) (*N* > 6) (**B**) quantification of apoptosis (apoptotic cysts/ovary) (*N* > 6) and (**C**) rate of vitellogenesis (>S8/<S8) (*N* > 12) in *D. sechellia* (14021–0248.25) flies fed a non-supplemented (–) standard diet (sd) or a standard diet supplemented with L-3,4-dihydroxyphenylalanine (1 mg/ml, L-DOPA); dopamine (1 mg/ml, DA); tyramine (2 mg/ml, TA) or octopamine (2 mg/ml, OA); or a non- pretreated morinda diet (md) or a morinda diet pre-treated with catechol-O-methyltransferase (2.5 U per gram of fruit, md + COMT). Different letters denote significant differences (p < 0.01) using ANOVA followed by Tukey's test. Error bars represent s.e.m. (**D**) The total ion chromatogram (top trace) shows all compounds present in morinda extract, and the extracted ion chromatogram (lower trace) corresponds to the exact mass of sum formula of L-3,4-dihydroxyphenylalanine (L-DOPA) present in the fruit; as analysed by UHPLC-MS. (**E**) Egg production (eggs/female/day) (*N* > 3) in *D. sechellia* (14021–0248.25) flies fed a morinda diet (md) non-pre-treated (–), pre-treated with catechol-O-methyltransferase (2.5 U per gram of fruit, COMT) or α-methyl-DOPA (0.4 mM, mDp). Different letters denote significant differences (p < 0.01) using ANOVA followed by Tukey's test. Error bars represent s.e.m.

The following figure supplement is available for figure 2:

**Figure supplement 1**. *D. sechellia* female fertility is modulated by morinda.

of fruit pulp (*Figure 2D*) in morinda extracts analysed by ultra-high-performance liquid chromatography coupled to mass spectrometry (UHPLC-MS). Notably, we did not detect DA, OA or TA in morinda. Unripe (150.99 ± 8 ng L-DOPA [mean ± s.e.m; *N* = 3] per gram of fruit) and overripe (147.47 ± 8 ng L-DOPA [mean ± s.e.m; *N* = 3] per gram of fruit) morinda fruits contained L-DOPA in equal amounts, what suggests that oxidisation, known to occur commonly in monoamines through atmospherical exposure, is prevented in morinda. The preservation of L-DOPA is likely due to the high carboxylic acid content—octanoic and hexanoic acids have been shown to inhibit diphenol oxidase activity (*Guo et al., 2010*)—and the ensuing low pH of the fruit (4.1 ± 0.1 [*N* = 7] and 3.6 ± 0.1 [*N* = 5] [mean ± s.e.m.], ripe and overripe morinda, respectively; for comparison, 5.7 ± 0.2 [mean ± s.e.m.], [*N* = 4], ripe banana; p = 0.0001; one-way ANOVA), known for its antioxidant properties. Banana is reported to contain up to 1 mg DA per gram of fresh weight (*Kanazawa and Sakakibara, 2000*). Notably, DA in banana is synthesized directly from TA (*Deacon and Marsh, 1971*) and not via synthesis

of L-DOPA. Presumably, the absence of L-DOPA in banana renders this non-host fruit unable to stimulate egg production in *D. sechellia* (see *Figure 1B*).

Is the presence of L-DOPA in morinda then necessary for the stimulatory effect on egg production? To address this question, we depleted levels of L-DOPA in morinda by pre-incubating fruit pulp with catechol-O-methyltransferase (COMT, 2.5 U/g morinda), an enzyme that catabolises L-DOPA into an unusable product (*Gordonsmith et al., 1982*). Treatment with COMT suppressed the anti-apoptotic effects of morinda (*Figure 2B*) and hindered the fruit from stimulating vitellogenesis (*Figure 2— figure supplement 1*) and egg production in *D. sechellia* (*Figure 2E*), reducing the number of eggs to levels in a standard diet (compare *Figure 2A* and *Figure 2E*). Additionally, we added α-methyl-DOPA (mDp, 0.4 mM), a non-hydrolysable L-DOPA analogue and a competitive inhibitor of the enzyme dopa decarboxylase that converts L-DOPA into DA, separately to the fruit. The presence of mDp in morinda strongly reduced egg production (*Figure 2E*); feeding a corresponding mDp-supplemented diet to *D. melanogaster* wild-type Berlin did not have such an effect (not shown). Notably, mDp in the fruit did not hinder oogenesis; instead, eggs were retained in the ovaries (*Figure 2—figure supplement 1*). This halt in ovulation was completely reversed one day after flies were moved to fresh medium (*Figure 2—figure supplement 1*), and, remarkably, oviposition was significantly enhanced when morinda was offered as an oviposition substrate (*Figure 2—figure supplement 1*). We conclude that morinda contains L-DOPA and that its presence in the fruit pulp is both sufficient and necessary to stimulate egg production in *D. sechellia*.

Morinda, accordingly, provides *D. sechellia* with the DA precursor necessary for the progression of oogenesis. Although providing a critical chemical, the acidity of the fruit, which helps preserve L-DOPA by preventing oxidisation, creates a hostile environment for the eggs to develop in. How do flies ensure that the eggs survive in this toxic environment? An interesting observation provides one clue. Eggs of *D. sechellia* are characteristically large compared to those of sibling species *D. melanogaster*, *D. simulans*, *Drosophila ananassae*, *Drosophila erecta*, *Drosophila mojavensis*, *Drosophila persimilis*, *Drosophila pseudoobscura*, *Drosophila virilis*, *Drosophila willistoni* and *Drosophila yakuba* (*Markow et al., 2009*). In accordance, we observed *D. sechellia* eggs to be 45% larger in size compared to eggs of *D. melanogaster* wild-type Berlin (*Figure 3A*), in a standard diet condition. Upon being fed a morinda diet, eggs of *D. sechellia* increased in volume twofold compared to eggs of conspecifics fed standard medium (*Figure 3A* and *Figure 3—figure supplement 1*); the resulting eggs had an almost threefold larger volume than did *D. melanogaster* wild-type Berlin eggs in standard conditions (*Figure 3A*). This effect could be replicated by supplementing standard diet with L-DOPA or, notably, DA (*Figure 3B*). On the other hand, adding COMT or mDp to a morinda diet prevented the fruit from having any effect on the volume of eggs (*Figure 3C*). These results indicate that egg size is also under a dopaminergic control regime during egg development. The increased size may provide buffering capacity to the eggs, helping them to cope with the toxic environment of the host. Indeed, the hatching rate of eggs transferred onto a morinda-containing medium was significantly higher in the enlarged eggs of flies fed morinda than in the smaller eggs of individuals fed only standard medium (*Figure 3D*). The low number of ovarioles of *D. sechellia* (*R'kha et al., 1997*) thus seems to be a result of a trade-off between number and size (*Figure 3—figure supplement 2*), favouring the enlarged eggs of females fed morinda.

Genetic changes in a single gene of the steroid hormone biosynthetic pathway made *Drosophila pachea* dependent on the uncommon sterols of its host plant, the toxic senita cactus (*Lang et al., 2012*). Likewise, the requirement of dietary L-DOPA suggests that its metabolism is impaired in *D. sechellia*. We next quantified L-DOPA in fly-tissue extracts by UHPLC-MS. *D. sechellia*—kept on standard diet—showed significantly lower levels of L-DOPA in whole flies, bodies and ovaries than did *D. melanogaster* wild-type Berlin (*Figure 4A*). However, in *D. sechellia* fed a morinda diet, L-DOPA levels were significantly increased compared to levels in conspecifics fed a standard diet, and surpassing those of *D. melanogaster* wild-type Berlin in standard diet (*Figure 4B*). In short, these results demonstrate that under laboratory conditions, L-DOPA is greatly reduced in *D. sechellia*, and that this L-DOPA deficiency can be remedied by a diet supplemented with morinda fruit.

The oxidation of the precursor amino acid tyrosine into L-DOPA is the first and rate-limiting step in the DA biosynthetic pathway (*Levitt et al., 1965*). This oxidation is catalysed by the enzyme tyrosine hydroxylase (TH), which is encoded by the *pale* (*ple*) locus in *Drosophila* (*Neckameyer and White, 1993*). The whole-fly expression of TH-PLE was increased in *D. sechellia* relative to in *D. melanogaster* wild-type Berlin (*Figure 4C*) and Canton-S (*Figure 4—figure supplement 1*), as revealed by Western

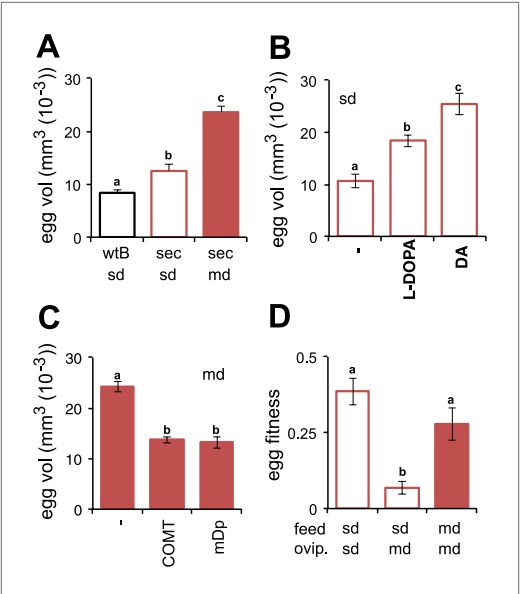

**Figure 3.** Morinda enhances early fitness. (**A–C**) Volume (mm³(10⁻³)) of *D. melanogaster* wild-type Berlin (wtB) and *D. sechellia* (14021–0248.25, sec) eggs produced by flies fed a standard diet (sd) or morinda diet (md) (*N* > 15) (**A**); *D. sechellia* (14021–0248.25) flies fed a non-supplemented (−) standard diet (sd), or supplemented with L-3,4-dihydroxyphenylalanine (1 mg/ml, L-DOPA) or dopamine (1 mg/ml, DA) (*N* > 23) (**B**); *D. sechellia* (14021–0248.25) flies fed a non-pre-treated (−) morinda diet (md), or pre-treated with catechol-O-methyltransferase (2.5 U per gram of fruit, COMT) or α-methyl-DOPA (0.4 mM, mDp) (*N* > 10) (**C**). (**D**) Egg hatching rate (*N* > 5) in *D. sechellia* (14021–0248.25) fed (feed) a standard diet (sd) or morinda diet (md), ovipositing (ovip.) in either media. Different letters denote significant differences (p < 0.01) using ANOVA followed by Tukey's test. Error bars represent s.e.m.

The following figure supplements are available for figure 3:

**Figure supplement 1**. Volume of *D. sechellia* eggs is modulated by morinda diet.

**Figure supplement 2**. Female resource investment on fertility is conserved in *D. sechellia*.

blots. Thus, L-DOPA impairment in *D. sechellia* seems not to result from a low expression of its synthesizing enzyme. To assess if the activity of TH-PLE was altered in *D. sechellia*, we compared the ratios of product (L-DOPA and DA) to substrate (tyrosine) in whole-fly extracts. *D. sechellia* showed a 3.3-fold decrease in the L-DOPA/tyrosine ratio as compared to in *D. melanogaster* wild-type Berlin (**Figure 4D**) with a 1.4-fold increase in tyrosine content (**Figure 4—figure supplement 2**). However, in *D. sechellia* the DA/tyrosine ratio was increased 2.4 times with respect to in *D. melanogaster* wild-type Berlin (**Figure 4D**). DA itself was increased 4.8 times in *D. sechellia* (**Figure 4F**). In sum, we conclude that TH-PLE is active; in fact, DA levels are drastically enhanced in *D. sechellia*.

TH-PLE activity is negatively regulated via direct physical interactions with the protein Catecholamines up (Catsup) (**Stathakis et al., 1999**). Interestingly, *Catsup* loss-of-function mutations cause hyperactivation of TH-PLE and abnormally high levels of catecholamines (**Stathakis et al., 1999**), as well as infertility due to maternal arrest of oogenesis (**Schupbach and Wieschaus, 1991**; **Stathakis et al., 1999**). These phenotypes prompted us to investigate whether Catsup is impaired in *D. sechellia*. We cloned the ortholog of *Catsup* from *D. sechellia*, which revealed a 45 bp in-frame deletion of 15 amino acids in a predicted zinc-binding region of the protein (**O'Donnell et al., 2002**) and seven single amino acid exchanges (**Figure 4E**). Additionally, Western blots of whole-fly extracts revealed 2- to 3.7-fold decrease in CATSUP expression in *D. sechellia* compared to CATSUP expression in *D. melanogaster* wild-type Berlin (**Figure 4C**) and Canton-S (**Figure 4—figure supplement 1**). The reduced expression of *D. sechellia* CATSUP would explain the high DA levels in *D. sechellia* (see **Figure 4F**).

Is *Catsup* then responsible for the reproductive phenotypes in *D. sechellia*? An allele (In270Del) similar to *D. sechellia Catsup* has been described for a natural *D. melanogaster* population (DGRP-357) (**Carbone et al., 2006**) (**Figure 4E**). *Catsup^In270Del* has been associated with polymorphisms in the number of sensory bristles, starvation resistance and locomotor behavior (**Carbone et al., 2006**). As for *D. sechellia*, we found increased DA levels in adult *D. melanogaster* DGRP-357 compared to in *D. melanogaster* wild-type Berlin (**Figure 4F**). Reciprocally, *D. sechellia* showed an increased number of sensory bristles that were present in flies of three geographically different populations (**Figure 4—figure supplement 3**). To test if *Catsup^In270Del* was sufficient to generate egg phenotypes on par with those of *D. sechellia*, we next examined egg growth in *D. melanogaster* DGRP-357 females. Both *D. sechellia* traits—low levels of vitellogenesis (**Figure 4G**) and enlarged eggs (**Figure 4H**)—were present in *D. melanogaster* DGRP-357. Furthermore, *D. melanogaster* DGRP-357 produced fewer eggs than did *D. melanogaster* wild-type Berlin (**Figure 4I**), and the ovariole rate of egg production in *D. melanogaster* DGRP-357 females did not differ significantly from that of *D. sechellia* (p = 0.2104; Student's *t* test). To control

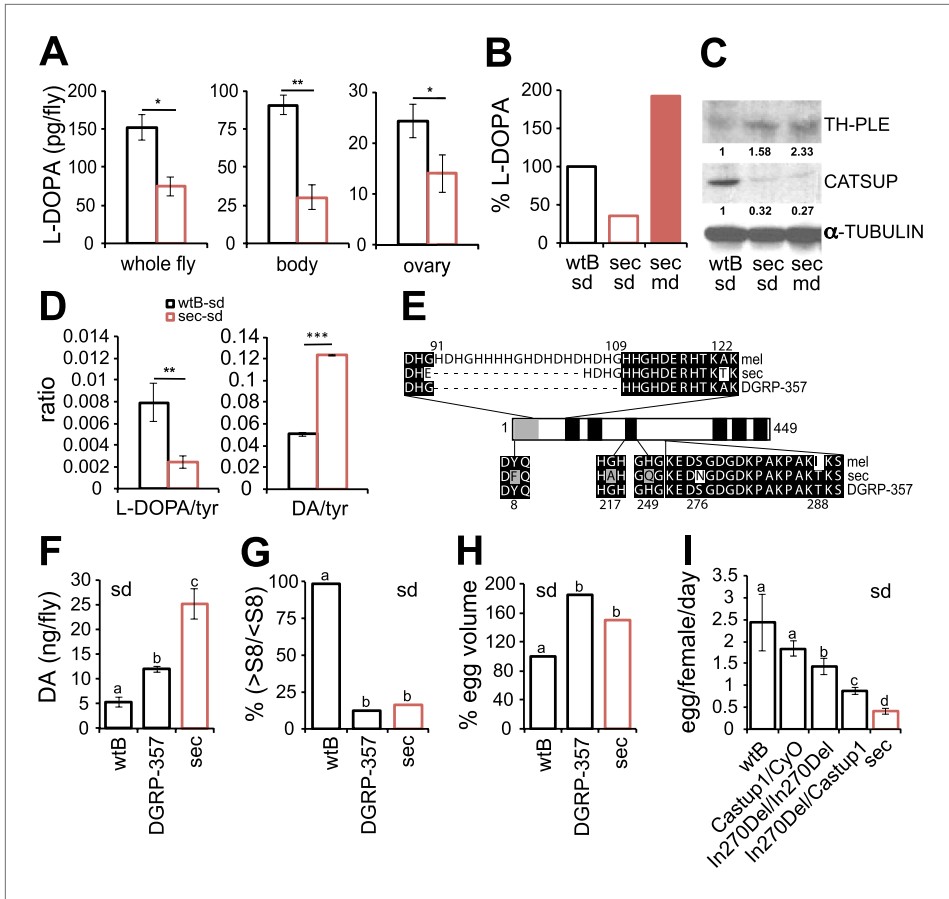

**Figure 4**. Dopamine metabolism is impaired in *D. sechellia*. (**A**) L-3,4-dihydroxyphenylalanine quantification (L-DOPA pg/fly) (*N* = 3) in whole fly, bodies and ovaries of female *D. melanogaster* wild-type Berlin (wtB) and *D. sechellia* (14021–0248.25, sec) fed a standard diet (sd). *p < 0.05 and **p < 0.002 using Student's *t* test. (**B**) Relative L-3,4-dihydroxyphenylalanine (% pg L-DOPA per mg body, % L-DOPA) (*N* = 3) in female *D. melanogaster* wild-type Berlin (wtB) and *D. sechellia* (14021–0248.25, sec) fed a standard diet (sd) or morinda diet (md). p = 0.0062 and p = 0.018 using Student's *t* test *D. melanogaster* vs *D. sechellia* fed, respectively, a standard diet or morinda diet. (**C**) Western blots of total protein whole-fly extracts for TH-PLE, CATSUP, and α-TUBULIN as a loading control, in *D. melanogaster* wild-type Berlin (wtB) and *D. sechellia* (14021–0248.25, sec) fed a standard diet (sd) or morinda diet (md). The numbers under TH-PLE and CATSUP protein lanes indicate the relative protein levels (normalised to α-TUBULIN). (**D**) Ratios of L-3,4-dihydroxyphenylalanine (L-DOPA/tyr) (*N* = 3) and dopamine (DA/tyr) (*N* = 3) to tyrosine substrate in female *D. melanogaster* wild-type Berlin (wtB) and *D. sechellia* (14021–0248.25, sec) fed a standard diet (sd). **p < 0.007 and ***p < 0.000007 using Student's *t* test. (**E**) Drosophila CATSUP protein structure scheme showing a signal peptide (grey box) and six trans-membrane domains (black boxes). Deletions (dash) and exchanges (grey or white) of amino acids in *D. sechellia* (14021–0248.25, sec) compared to in *D. melanogaster* wild-type Berlin (wtB) and in *D. melanogaster* DGRP-357 (DGRP-357) CATSUP are indicated. (**F**–**H**) Dopamine (DA ng/fly) (*N* = 3) (**F**), relative rate of vitellogenesis (% >S8/<S8) (*N* > 10) (**G**), and egg-volume (% mm³(10⁻³)) (n > 10) (**H**) in *D. sechellia* (14021–0248.25, sec), *D. melanogaster* wild-type Berlin (wtB) and *D. melanogaster* DGRP-357 (DGRP-357) fed a standard diet (sd). (**I**) Egg production (egg/female/day) (*N* > 3) in *D. melanogaster* wild-type Berlin (wtB), heterozygote flies (*Catsup¹/CyO*), *D. melanogaster* DGRP-357 (*Catsup^In270Del/Catsup^In270Del*), trans heterozygote flies (*Catsup^In270Del/Catsup¹*) and *D. sechellia* (14021–0248.25, sec), fed a standard diet (sd). Different letters denote significant differences (p < 0.05) using ANOVA followed by Tukey's test (**F**–**I**). Error bars represent s.e.m.

The following figure supplements are available for figure 4:

**Figure supplement 1**. TH-PLE and CATSUP expression in Drosophila.

**Figure supplement 2**. Tyrosine quantification in Drosophila.

*Figure 4. Continued on next page*

*Figure 4. Continued*

**Figure supplement 3**. Sensory bristles.

**Figure supplement 4**. L-DOPA rescues diminished *Catsup^{In270Del}/Catsup^1* egg production.

**Figure supplement 5**. Conserved *Catsup* sequence in *D. sechellia*.

for non-identified gene mutations present in *D. melanogaster* DGRP-357 that could contribute to lowering egg production, we generated heteroallelic flies carrying *Catsup^{In270Del}* in trans-heterozygosis with the null *Catsup^1* allele (*Stathakis et al., 1999*). *Catsup^{In270Del}/Catsup^1* females displayed significantly decreased egg production, below that of *D. melanogaster* wild-type Berlin and parental lines (*Figure 4I*), that could be rescued by feeding the flies a standard diet supplemented with L-DOPA (*Figure 4—figure supplement 4*). In sum, these results show that *D. melanogaster* flies carrying a *D. sechellia Catsup*-like allele mirror *D. sechellia's* low production of eggs.

The formation of TH-PLE functional protein begins during the maternal phase of egg development (*Pendleton et al., 2007*). Pharmacologically inhibiting TH-PLE arrests oocytes at S8 stage of development, which is prevented by the co-administration of L-DOPA (*Pendleton et al., 1996*). We thus asked if the diet requirement of morinda for *D. sechellia* oocyte progression results from a maternal impairment of the enzyme synthesizing L-DOPA. Ple mRNA has been shown expressed by in situ hybridization in nurse- and follicle cells (*Neckameyer, 1996*). We detected TH-PLE in the ooplasm of *D. melanogaster* wild-type Berlin developing oocytes (*Figure 5A*). Remarkably, *D. sechellia* oocytes showed big masses of TH-PLE abnormally accumulated in their ooplasm (*Figure 5B*), suggesting that the nurse-to-oocyte traffic of maternal Ple is hindered in this fly.

*Catsup* encodes the *Drosophila* ortholog of the mammalian ZIP7/SLC39A7 zinc transporter and has recently been associated with trafficking of proteins during development (*Groth et al., 2013*). Flies carrying the I288T amino acid exchange shared in *D. sechellia* CATSUP and *D. melanogaster* DGRP-357 CATSUP (see *Figure 4E*) accumulate abnormal amounts of proteins in developing tissue (*Groth et al., 2013*). Notably, as for *D. sechellia*, we found big masses of TH-PLE abnormally accumulated in the ooplasm of *D. melanogaster* DGRP-357 oocytes (*Figure 5C*) strongly suggesting that CATSUP could regulate the nurse-to-oocyte traffic of maternal Ple in *Drosophila*. CATSUP was expressed in the perinuclear endoplasmic reticulum- and golgi-like cisternae of the nurse cells of S8 oocyte cysts (*Figure 5D*), what results a suitable intracellular distribution for a protein with traffic nurse-to-oocyte function. We found, additionally, CATSUP expressed in the spermathecal secretory cells (SSC) of the female reproductive tract (*Figure 5G*). As for whole flies extracts, *D. sechellia* and DGRP-357 showed diminished expression of CATSUP in the nurse cells (*Figure 4E,F*) and SSC (*Figure 4H,I*). A recent study showed that the SSC require a not-yet identified secretory pathway to control the ovulation and downstream movement of the eggs (*Sun and Spradling, 2013*). The ablation of the SSC causes ovoviviparity in *D. melanogaster* (*Schnakenberg et al., 2011*), a condition naturally occurring in *D. sechellia* (*Markow et al., 2009*). Remarkably, 81.8% of *D. melanogaster* DGRP-357 mated females had an egg in their uterus (against 22.7% of *D. melanogaster* wild-type Berlin; p < 0.0001, chi-squared test); many eggs showed an advanced embryonic development and even hatched inside the female reproductive tract (*Figure 5—figure supplement 1*).

Null *ple* alleles (*Neckameyer and White, 1993*) and conditional *ple* mutants at its restrictive temperature (*Pendleton et al., 2007*) are lethal at late embryonic stage. Zygotic *ple* expression starts at later stages of embryonic development (*Lundell and Hirsh, 1994*; *Pendleton et al., 2007*), with early stages depending on maternal TH-PLE. Maximal survival of *D. sechellia* egg (*R'Kha et al., 1991*) matches the proportion of eggs hatching in the term of 2.5 hr after oviposition (*Markow et al., 2009*)—that is: laid at ~22 hr of embryonic development—what coincides with the peak of DA synthesis during embryogenesis (*Wright, 1987*). Thus, being ovoviviparous (*Markow et al., 2009*) may benefit *D. sechellia* embryos with dietary maternal L-DOPA and, additionally, provide the eggs with a protective cuticle (secreted at a late stage of embryonic development) highly resistant to environmental hazards. In fact, susceptible siblings embryos resist contact with morinda at late developmental stages (*R'Kha et al., 1991*). While no *D. melanogaster* (N = 345) eggs or *D. sechellia* sister species *D. simulans* (N = 58) and *D. mauritiana* (N = 226) eggs completely hatched in morinda pulp,

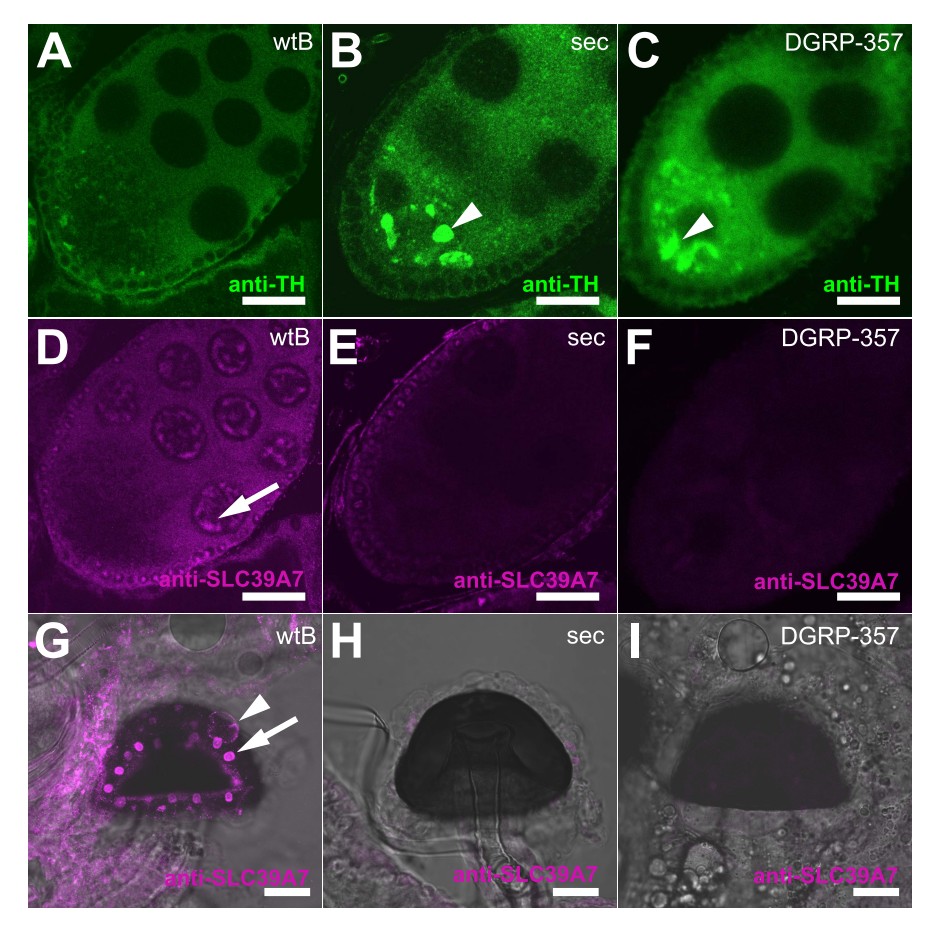

**Figure 5**. Expression of TH-PLE and CATSUP in Drosophila female reproductive system. (**A**–**C**) Confocal images showing striking accumulation (arrowhead) of TH-PLE (green, anti-TH) in *D. sechellia* (14021–0248.25) (**B**) and *D. melanogaster* DGRP-357 (**C**), compared to in *D. melanogaster* wild-type Berlin (wtB) (**A**) oocytes. (**D**–**F**) Confocal images showing CATSUP (magenta, anti-SLC39A7) expressed in the nurse cells (arrow) of *D. melanogaster* wild-type Berlin (wtB) oocytes (**D**) and absent in the nurse cells of *D. sechellia* (14021–0248.25, sec) (**E**) and *D. melanogaster* DGRP-357 (DGRP-357) (**F**) oocytes. (**G**–**I**) Confocal image showing CATSUP expressed in the nuclei (arrow) and the membrane (arrowhead) of *D. melanogaster* wild-type Berlin (wtB) spermatheca secretory cells (**G**), and absent from *D. sechellia* (14021–0248.25) (**H**) and *D. melanogaster* DGRP-357 (DGRP-357) (**I**). Scale bar 20 μm.

The following figure supplement is available for figure 5:

**Figure supplement 1**. Egg hatching in morinda.

33% of *D. melanogaster* eggs produced by flies carrying *Catsup^{In270Del}* allele DGRP- (*N* = 271) hatched to larva 1 (*Figure 5—figure supplement 1*), some even reaching larva 2 and larva3. These results illustrate how a mutation that reduces female fertility can concomitantly contribute to increase fecundity in a novel niche.

How do *D. sechellia* adults cope with the toxic environment of morinda fruits to lay their eggs in the first place? In non-*sechellia* Drosophilids, morinda acids toxicity is mostly manifested as locomotion disorders followed by slowness in movement until complete immobilization and death (*Legal et al., 1992*). These behavioral phenotypes resemble the neurotoxic effects described in *Drosophila* upon exposure to fungal-derived organic volatiles of similar chemical structure to morinda carboxylic acids (*Inamdar and Bennett, 2014*), which have been portrayed as reducing DA in the flies (*Inamdar et al., 2013*). Thus, we asked if incrementing DA could aid the adult flies in overcoming the behavioral effects of morinda carboxylic acids. To test this, we fed adult *D. melanogaster* wild-type Berlin a

synthetic diet supplemented with increasingly amounts of DA and tested their survival upon exposure to morinda carboxylic acids. After a few minutes of experiencing a natural concentration mix of octanoic and hexanoic acids, *D. melanogaster* wild-type Berlin flies fed a standard diet showed general restlessness and altered equilibrium with incorrect positioning of tarsi, followed by slowness in movement until complete immobilization. A 66% of females (*Figure 6A*) and 53% of males (*Figure 6B*) were completely immobilized after 120 min of exposure to the mix of acids. Dietary administration of DA declined the locomotion behavioral effects of morinda carboxylic acids in a dose-dependent manner (*Figure 6A,B*). *D. melanogaster* wild-type Berlin flies (females and males) treated with 100 mg/ml DA reached a survival degree not significantly different from that of *D. sechellia* (*Figure 6C,D*). These results suggest that the endogenously increased DA levels in *D. sechellia* (see *Figure 4F*) contribute to override the behavioral effects of morinda carboxylic acids. On the other hand, DGRP-357 flies—of intermediate DA levels (see *Figure 4F*)—showed *D. melanogaster* levels of resistance to morinda ($41.2 \pm 4.7$ and $52.5 \pm 2.8\%$ survival in morinda, respectively in DGRP-357 [$N = 6$] and *D. melanogaster* wild-type Berlin [$N = 4$]; $p = 0.09$ using Student's $t$ test), suggesting that further genes might contribute to elevate the levels of DA in *D. sechellia*. *D. sechellia* shows protein levels of TH-PLE, the rate limiting enzyme in DA production, higher than in the sister species *D. simulans* and *D. mauritiana*, and

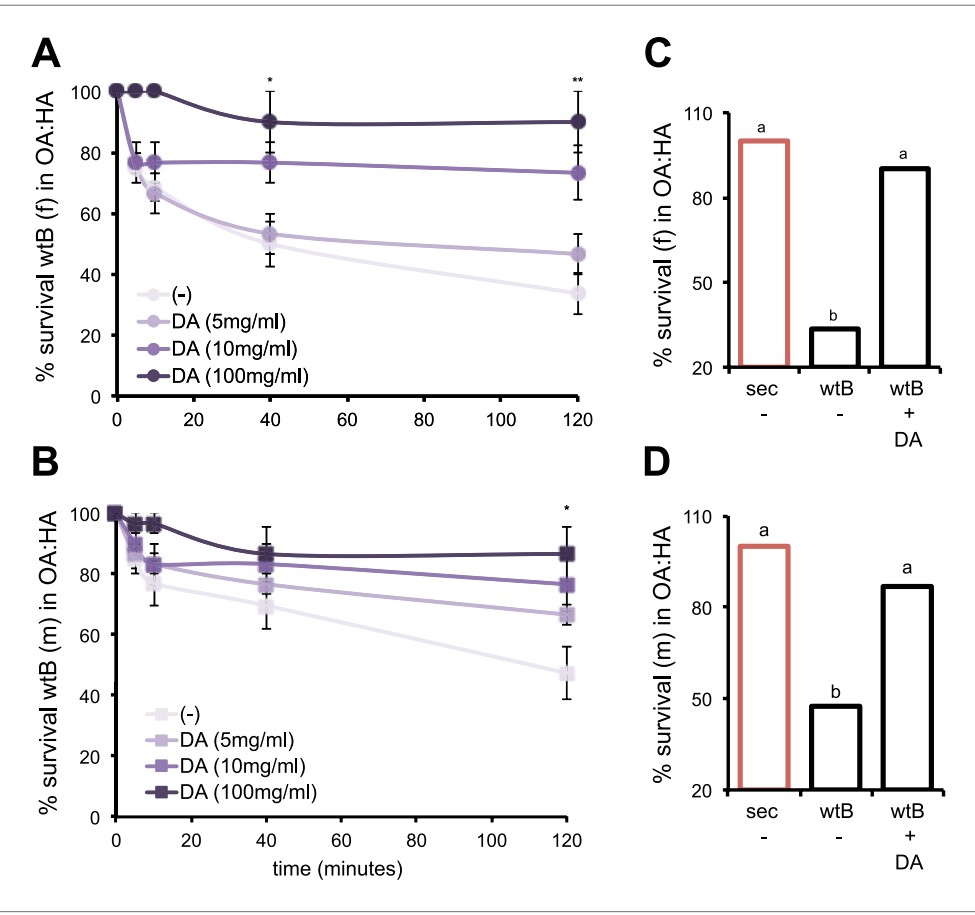

**Figure 6**. DA contributes to the behavioral resistance to morinda carboxylic acids. (**A–B**) Survival kinetic curves for *D. melanogaster* wild-type Berlin (wtB) (**A**) females (f, $N > 3$) and (**B**) males (m, $N > 3$) exposed to morinda carboxylic acids (OA:HA) and fed a standard diet supplemented with either no (−), or increasing doses of DA (5 mg/ml, 10 mg/ml and 100 mg/ml). *$p < 0.02$ and **$p < 0.002$ using Student's $t$ test. (**C–D**) Survival (%) ($N > 3$) of *D. sechellia* (14021–0248.25, sec) and *D. melanogaster* wild-type Berlin (wtB) (**C**) females (f) and (**D**) males (m), fed a non-supplemented (−) or DA (+DA, 100 mg/ml) supplemented standard diet upon 2 hr exposure to morinda carboxylic acids (OA:HA). Different letters denote significant differences ($p < 0.01$) using ANOVA followed by Tukey's test. Error bars represent s.e.m.

in wild type *D. melanogaster* (wild-type Berlin, CS) and DGRP-357 (see *Figure 4E* and *Figure 4—figure supplement 1*). Additionally, in a screen for genes that differ in expression between *D. sechellia* and four geographically distinct populations of its generalist sister species *D. simulans*, *Wurmser et al. (2011)* showed lower expression of the DA catabolic enzyme (*Dopamine-N-acetyltransferase*), what probably contributes to almost quintuplicate DA in *D. sechellia* (see *Figure 4F*). Thus, the enhancement of DA could have served as a founder event in the history of morinda becoming an obligate host for the specialist *D. sechellia*.

Our results are compatible with an evolutionary scenario in which an original *Catsup* allele carrying a 45-bp deletion and six non-synonymous mutations was present in the ancestor of present-day *D. sechellia* (*Figure 4—figure supplement 5*), probably showing differentially low expression of CATSUP compared to *D. melanogaster* and the sister species *D. simulans* and *D. mauritiana* (see *Figure 4E* and *Figure 4—figure supplement 1*); what caused diminished egg production and concomitantly enhanced early survival in the fruit. A second event, further elevating DA (see *Figure 4F*), would have redeemed adult *D. sechellia* from the toxic effects of morinda acids allowing them to oviposit in ripe fruits, benefiting from morinda stimulating effect on egg production and from the lack of siblings competition, rendering *D. sechellia* dependent on morinda. This dependency would have, in turn, shaped the super specialization of the chemosensory system (*Dekker et al., 2006*; *Matsuo et al., 2007*; *Dworkin and Jones, 2009*) present in today's *D. sechellia* morinda obligate specialist.

## Conclusions

The molecular traits underlying adaptations and endurance to new toxic-hosts remain unknown. We present a novel role for the *Drosophila* catecholamine regulatory protein Catsup in maternal secretory functions and suggest that the malfunction of CATSUP contributed to *D. sechellia* becoming an obligate specialist on its toxic host. We propose an evolutionary scenario in which an initial mutation in *D. sechellia* DA metabolism caused impaired female fertility and fecundity, but concomitantly provided eggs and adults with resistance to morinda toxic acids. Together with an early loss of repellence (*Matsuo et al., 2007*), this initial maternally inherited tolerance allowed individuals to develop in morinda and feed the DA precursor, particularly enriched in this fruit, which strongly increased adult female fertility. In turn, the lack of repellence (*Matsuo et al., 2007*), and instead the strong attraction to morinda volatiles (*Dekker et al., 2006*), combined with the beneficial morinda effect on ovulation and egg laying (*Figure 2—figure supplement 1*), shaped *D. sechellia*'s preference to oviposit in its natural host (*Amlou et al., 1998*).

## Materials and methods

### Chemicals

All used chemicals were of commercial origin and used without further purification. UHPLC-MS- grade water and methanol were used for chromatography.

### Fly stocks and rearing

*D. sechellia* (14021–0248.03, 14021–0248.07, 14021–0248.08, 14021–0248.25, 14021–0248.27, 14021–0248.28, 14021–0248.31) were obtained from the *Drosophila* Species Stock Center (DSSC, https://stockcenter.ucsd.edu). *D. sechellia* (14021–0248.25), *D. simulans* (14021–0251.004) and *D. mauritiana* (14021–0241.01) were kindly provided by the Division of Chemical Ecology, Swedish University of Agricultural Sciences. *D. melanogaster* line wild-type Berlin was kindly provided by Silke Sachse. *D. melanogaster* line Oregon-R was kindly provided by Rafael Cantera. *D. melanogaster* pforta was captured in the Weingut Kloster Pforta (Naumburg, Germany). *D. melanogaster* lines Canton-S (BL1), DGRP-357 (BL25184), DGRP-437 (25194), DGRP-304 (BL25177) and *Catsup¹/CyO* (BL5138) were obtained at the Bloomington Stock Center at Indiana University (http://flystocks.bio.indiana.edu/). The *Catsup^{In270De}/Catsup¹* flies were selected as not carrying *CyO* from the F1 of a corresponding crossing of the parental lines DGRP-357 and *Catsup¹/CyO*. Flies were reared in vials (50 mm diameter × 95 mm high) at 25°C, 70% RH in L:D 12:12 on standard cornmeal–yeast–agar medium (standard diet), supplemented as specified in the text, or on fresh morinda pulp (morinda diet) collected from fruits of *M. citrifolia* plants kept in our greenhouse, or on banana. We observed no differences in survival to morinda fruit in our *D. sechellia* experimental stock along the successive generations (5 days survival in ripe morinda: 70.7 ± 5.6% and 83.2 ± 3.6% for new and old females [$N = 3$], p = 0.147 using Student's *t* test to compare stocks, respectively; 83.3 ± 16.7 and 64.7 ± 18.2 for new and old males, [$N = 3$], p = 0.493 using Student's *t* test to compare stocks, receptively), discarding an artificial adaptation to morinda acids. On the other hand,

morinda toxicity (*Legal and Plawecki, 1995*) hindered a permanent experimental stock of *D. melanogaster* in the fruit, for what flies were fed morinda for as long as the experiment lasted. Morinda carboxylic acids were added to the standard diet or agar plates as indicated in the text, using a range of natural concentrations (0.07% vol/V of 3:1 octanoic:hexanoic mix) (*Legal et al., 1994*).

## Adult survival

Triplicates of 10–20 flies were placed in vials (25 mm diameter × 95 mm high) containing ripe morinda and maintained at 25°C, 70% RH in L:D 12:12. Live adults were flipped daily to new vials, recording female and male survival. Data is expressed as percentage of females and males alive.

## Egg production and oviposition

Egg production was scored in groups of 10–20 females and males kept during 4 days in cages (37.5 mm diameter and 58 mm high) holding agar plates containing 5% sucrose and devoid of yeast at 25°C, 70% RH in L:D 12:12. Agar plates were changed daily and the total number of eggs was summed. Egg production is expressed as number of eggs per female per day, averaged over >5 independent experiments. Oviposition of the eggs retained by mated females fed a morinda diet supplemented with α-methyl-DOPA (0.4 mM, mDp) was scored in groups of 20 females transferred to non-supplemented standard diet or morinda diet as oviposition substrates. The number of laid eggs was averaged over three independent experiments and expressed as number of eggs per female per day.

## Ovariole staining and immunofluorescence

Ovaries ($N > 8$) were dissected and fixed in 500 µl of 4% paraformaldehyde in PT (phosphate buffer saline [PBS], 0.1% Triton X-100 [Sigma, St. Louis, MO]) during 30 min at room temperature. After being rinsed three times during 10 min in 1 ml PT, ovaries were incubated in 500 µl PT + 0.5 µl TOTO-3 (1 mM, Invitrogen, Life Technologies GmbH, Germany) + 0.5 µl SYTOX Orange (5 mM, Invitrogen, Life Technologies GmbH, Germany) and protected from light, overnight (ON) at 4°C. Ovaries were rinsed three times during 10 min in 1 ml PT and mounted in Vectashield (Sigma, St. Louis, MO). Oocyte-cyst stages were visually determined by inspection of stained ovaries under fluorescent microscope. Alternatively, fixed ovaries were blocked in PBST (PT, 5% normal goat serum [Sigma, St. Louis, MO]), incubated 24 hr at 4°C with primary antibody anti-TH (1:100, mouse, ImmunoStar, Acris Antibodies, GmbH, Germany) or anti-SLC39A7 (1:1000, rabbit, Sigma, St. Louis, MO), diluted in PBST, further rinsed three times in 1 ml PT during 10 min at room temperature (RT) and incubated respectively with fluorescently conjugated secondary antibodies anti-mouse (AlexaFluor488, Invitrogen, Life Technologies GmbH, Germany) and anti-rabbit (AlexaFluor546, Invitrogen, Life Technologies GmbH, Germany), diluted 1:250 in PBST at RT, protected from light. Ovaries were rinsed three times in 1 ml PT during 10 min and mounted in Vectashield (Sigma, St. Louis, MO). Confocal images were obtained at 1-µm intervals over 20 µm Z-stack using a LSM510 Meta confocal microscope (Zeiss, Jena, Germany).

## Apoptosis

Apoptosis in *D. sechellia* ovaries ($N > 6$) was visualized following the protocol by Arama and Steller (*McCall, 2004*). Briefly, live ovaries were dissected in PBS and incubated in a freshly prepared solution of 0.6 µg/ml acridine orange (Sigma, St. Louis, MO) for 5 min, at RT. Ovaries were rinsed briefly in PBS and mounted in a drop of Halocarbon 700 oil (Sigma, St. Louis, MO) and observed immediately. Confocal images were obtained at 1-µm intervals over 20 µm Z-stack using a LSM510 Meta confocal microscope (Zeiss, Jena, Germany).

## Feeding assay

Feeding assay was performed as described previously (*Riemensperger et al., 2011*). 10 *D. sechellia* 5-day-old flies (five females and five males) were starved for 2 hr at 25°C and transferred to vials with cornmeal-yeast food (standard diet) or morinda fresh pulp (morinda diet), containing 10 mM sulforhodamine B (Sigma, St. Louis, MO). Flies were allowed to feed for 1 hr and were immediately frozen at −20°C for 2 hr. Females and males were processed separately. Heads were removed to prevent contamination with eye pigments, and the bodies were homogenized in 250 µl of PBS. Samples were micro-centrifuged at 17,000×*g* (Eppendorf Centrifuge 5415 R) for 7 min at 4°C, after which the supernatant was collected, mixed with 60 µl of chloroform and micro-centrifuged for 6 min. The optical density of the supernatant was determined at 570 nm (BioSpectrophotometer, Eppendorf). Results are the mean of three independent determinations in each food condition.

## Amine and precursors quantification

Fresh morinda fruits (10 g) were homogenized in 15 ml 0.1 M perchloric acid, incubated during 5 min at RT and micro-centrifuged at 17,000×*g* (Eppendorf Centrifuge 5415 R) for 5 min at 4°C. The resulting supernatant was measured in a RS-3000-LTQ-Orbitrap XL instrument (Dionex and Thermo Fischer) (*Docimo et al., 2012*). L-3,4-dihydroxyphenylalanine (L-DOPA) values were estimated by calibration curve regression. Arithmetic mean and standard deviation were calculated over three independent samples and values expressed as nanogram of L-DOPA per gram of fruit. Other amines, DA, TA and OA, were not detected in morinda fruit under these conditions. Fly samples (5 whole flies, or 6 bodies or 10 ovaries) were processed in 50 µl of 0.1 M perchloric acid with 0.3 mM mDp as an internal standard, using ceramic beads (peqlab, Biotechnology GmbH) in TissueLyser LT (QIAGEN). Samples were micro-centrifuged twice at 17,000×*g* (Eppendorf Centrifuge 5415 R) for 5 min at 4°C and the supernatant measured as above. Tyrosine, L-DOPA and DA absolute values were calculated by calibration curve regression using XCMS/MZMatchR program (*Tautenhahn et al., 2008*). Arithmetic mean and standard deviation were calculated over three independent samples. For flies fed a standard diet, values were expressed as picogram of L-DOPA or DA per fly (fly weights showed no statistically significant difference [$p = 0.85564545$, *D. melanogaster* vs *D. sechellia*, Student's *t* test]). To compare flies fed different diets, values were expressed as picogram of L-DOPA per milligram of body tissue.

## Fruit pH measurements

The fruit pulp of ripe or overripe morinda and banana was mashed lightly and the pH was measured with a glass electrode. Results are the mean of three independent determinations for each fruit.

## Egg-hatching rate and size

*D. sechellia* 10–20 females and males fed a standard diet were placed in oviposition cages (37.5 mm diameter and 58 mm high) holding plates containing either a standard diet or fresh morinda pulp, at 25°C, 70% RH in L:D 12:12. In parallel, *D. sechellia* 10–20 females and males fed a diet of morinda were placed in oviposition cages (37.5 mm diameter and 58 mm high) holding plates containing fresh morinda pulp, at 25°C, 70% RH in L:D 12:12. Plates were changed every 0.5 hr to get pools of synchronized eggs and further incubated at 25°C. Hatching rate was expressed as the relative number of hatched eggs (larvae 1) to the total number of eggs produced. Egg size was measured within 1 hr of oviposition or on pre-fertilized eggs inside the ovary (there was not a statistically significant difference between these groups). For each species, the lengths (*L*) and widths (*W*) of 15–30 eggs were measured and their volumes were determined according to the formula *$(1/6)\pi W^2 L$* (*Markow et al., 2009*). Values were expressed as $mm^3$ ($10^{-3}$).

## Western blots

For each species, 20 adult flies were homogenized in 200 µl lysis buffer (50 mM Tris–HCl pH 7.5, 0.1% [vol/V] Triton X-100, 100 mM NaCl, 1 mM DTT, 10% glycerol, 15 mM EDTA) freshly prepared with protease inhibitor cocktail (Roche) added immediately before use, using ceramic beads (peqlab, Biotechnology GmbH, Germany) in TissueLyser LT (QIAGEN GmbH, Germany). Whole protein extracts of one same experiment were separated in parallel by 10% SDS-PAGE plus electronic transfer to PVDF membranes (BioRad, Germany). After being blocked in 5% non-fat milk in TBS-tween (TBS, 0.05% Tween-20 [Sigma, St. Louis, MO]) for 2 hr, at RT, membranes were incubated with 1:1000 dilutions of primary antibodies (anti-TH [mouse, ImmunoStar, Acris Antibodies, GmbH, Germany], or anti-SLC39A7 [rabbit, Sigma, St. Louis, MO] or anti-α-tubulin [mouse, Sigma, St. Louis, MO]) diluted in 2.5% non-fat milk in TBS tween ON at 4°C. Membranes were further washed in TBS-tween at RT and re-blocked in 10% non-fat milk TBS-tween for 10 min at RT before being incubated with 1:10000 dilution of corresponding secondary HRP-conjugated anti-mouse or anti-rabbit (BioRad, Germany). Proteins were detected using an enhanced chemiluminescence detection kit (Thermo scientific pierce, Germany). The densitometry of bands was performed using ImageJ package (http://imagej.nih.gov/ij/). Relative densities of TH-PLE and CATSUP were scaled using the relative densities of the loading-controls (α-Tubulin).

## Gene cloning and sequencing

For each species, we prepared total RNA (Trizol, Invitrogen, Life Technologies GmbH, Germany) and synthesized cDNA (SuperScript III First-Strand Synthesis System, Invitrogen, Life Technologies

GmbH, Germany) that was used to amplify *Catsup* transcript by PCR (Advantage HD Polymerase, Clontech) with specific primers (forward 5'-ATGGCCAAACAAGTGGCTGA-3' and reverse 5'-TTACTCGAACTTGGCGATAAC-3'). Each PCR product was cloned in pCRII vector (Invitrogen, Life Technologies GmbH, Germany) and at least 10 colonies were picked for plasmid DNA purification and sequencing. These sequences were aligned using MegAlign (DNASTAR) and a consensus sequence was obtained for each species. The sequence of *Catsup*[In270Del] was obtained from the *Drosophila* Polymorphism Database (DPDB) (*Casillas et al., 2005*).

### Sternopleural bristle quantification

Total (macro and micro) bristles of each sternopleural plaque were counted in females of *D. melanogaster* wild-type Berlin (*N* = 34), *D. melanogaster* Canton-S (*N* = 54), *D. melanogaster* DGRP-357 (*N* = 61), DGRP-437 (*N* = 18), DGRP-304 (*N* = 16)—three independent lines carrying *Catsup*[In270Del] allele—and *D. sechellia* lines 14021–0248.08 (*N* = 98), 14021–0248.25 (*N* = 74), 14021–0248.27 (*N* = 25), 14021–0248.28 (*N* = 56) and 14021–0248.31 (*N* = 50). Data is expressed as frequency histograms and a bar-graph showing the mode of each population.

### Behavioural assay

5-day-old flies were fed a standard diet either not- or supplemented with increasingly amounts of DA (5 mg/ml, 10 mg/ml and 100 mg/ml) during 16 hr prior to their behavioural assessment, at 25°C, 70% RH in L:D 12:12. The locomotion behaviour was scored in groups of 30 females and 30 males kept in cages (37.5 mm diameter and 58 mm high) holding agar plates containing 5% sucrose and a mix of 0.07% vol/V 3:1 octanoic:hexanoic acids. *M. citrifolia* fruits contain highly volatile compounds that confer toxicity (*Legal et al., 1994*), with the two most abundant being octanoic acid (58%) and hexanoic acid (19.24%) (*Farine et al., 1996*). Octanoic acid is the most toxic compound in the ripe fruit (*Legal et al., 1994*), which *D. sechellia* has been able to overcome by the development of both larval and adult tolerance mechanisms (*R'Kha et al., 1991*; *Legal et al., 1992*; *Amlou et al., 1997*, *1998*; *Jones, 1998*, *2001*, *2005*). Given these observations, the ratio provided on par. A drop of fresh yeast was added to the plates forcing the attracted flies to enter in contact to the acids. Flies were carefully observed under a stereoscope and the number of immobilized flies was recorded every 5 min independently for females and males. Results are the mean of three to five independent determinations in each food condition.

## Acknowledgements

We wish to thank Regina Stieber, Sandra Scholz and Céline Callens for expert technical support; and Emily Wheeler, Boston, for editorial assistance. Stocks obtained from the Bloomington *Drosophila* Stock Center (NIH P40OD018537) were used in this study. This work was supported by the Max Planck Society (SL-LL, AS, MK, MCS and BSH) and by grants from the ESPCI ParisTech and the Fondation de France to SB (TR and SB).

## Additional information

### Competing interests

BSH: Vice president of the Max Planck Society, one of the three founding funders of *eLife*, and a member of *eLife's* Board of Directors. The other authors declare that no competing interests exist.

### Funding

| Funder | Grant reference number | Author |
|---|---|---|
| Max-Planck-Gesellschaft | | Sofía Lavista-Llanos, Aleš Svatoš, Marco Kai, Marcus C Stensmyr, Bill S Hansson |
| Centre National de la Recherche Scientifique | ESPCI ParisTech | Thomas Riemensperger, Serge Birman |
| Fondation de France | | Thomas Riemensperger, Serge Birman |

The funders had no role in study design, data collection and interpretation, or the decision to submit the work for publication.

## Author contributions

SL-L, Conception and design, Acquisition of data, Analysis and interpretation of data, Drafting or revising the article; AS, MK, TR, Acquisition of data, Analysis and interpretation of data; SB, MCS, BSH, Conception and design, Drafting or revising the article

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
