## [Decision Letter]

Thank you for sending your work entitled “Dopamine drives *Drosophila sechellia* adaptation to its toxic host” for consideration at *eLife.* Your article has been favorably evaluated by K VijayRaghavan (Senior editor) and 3 reviewers, of whom Mani Ramaswami is a member of our Board of Reviewing Editors.

The following individuals responsible for the peer review of your submission have agreed to reveal their identity: Giovanni Bosco and John Jaenike.

The Reviewing editor and the other reviewers discussed their comments before we reached this decision, and the Reviewing editor has assembled the following comments to help you prepare a revised submission.

The paper identifies the genetic basis for the evolution of a feeding behavior in *Drosophila* species. It maps the mutation conferring preference for morinda fruit to mutations in catsup, a *Drosophila* catcholamine regulatory protein encoding gene, which affects dopamine synthesis as well as the presence of a dopamine precursor l-Dopa in morinda fruit. It further shows a role for Catsup in maternal secretory function.

The paper is on balance a very nice piece of work that is potentially suitable for publication in *eLife.* However, there are some experimental gaps in the paper, as well as additional experimental and/or informational details the need to be included to tighten and complete the study.

The essential revisions that we request are:

1) Please provide data to show that a morinda diet does not affect egg-laying in Drosophila melanogaster. Particularly as previous work shows that contact with the ripe fruit kills any member of the *D. melanogaster* complex with the exception of *D. sechellia*, which is attracted by and prefers it (Legal and Plawecki 1995).

2) Please provide DAPI visualization of the ovaries to either confirm or deny apoptosis as a possible culprit for the depressed oviposition in *D. sechellia* when on standard media. Alternatively, some other method of detecting apoptosis, like TUNEL or activated caspase staining could be used to determine whether an oogenesis checkpoint has been triggered. Subsequently, demonstrating that apoptosis is no longer present when the flies are raised on Morinda fruit is also necessary. This is required because stressors can cause egg chambers in the ovaries to be eliminated by apoptosis at oogenesis checkpoints in region-2/3 of the germarium or stage 7/8 egg chambers (the mid-oogenesis checkpoint) (Drummond-Barbosa 2001; McCall 2004), which could be a possible explanation for the deceased egg numbers.

3) Please test whether heteroallelic *D. melanogaster* fly lines carrying *Catsup^ln270del^* and the null *Catsup^1^* allele, show egg depression that can be rescued by feeding the flies l-DOPA (as with the *D. sechellia* experiments). This is a very important experiment that would complete the story.

4) The paper states that no *D. melanogaster* wild-type eggs hatched in Morinda pulp but 13.5% of DGRP-357 eggs do. Could this be a strain difference in *D. melanogaster* lines? This issue could be addressed by testing a few other *D. melanogaster* lines or by genetic rescue.

5) At the end of the Results and Discussion please present a clear and complete description of the evolutionary scenario by which *D. sechellia* first became tolerant to and then dependent on morinda fruit. Even if the scenario includes some speculations, it is important to see the whole thing put together.

Minor comments to which we request a constructive response:

1) The authors state that some *D. sechellia* in Figure 1 were raised on a Morinda diet (pulp from the fruit collected from the greenhouse). On its native islands, *D. sechellia* specializes on the ripe fruit of Morinda citrifolia (Louis and David 1986, Lachaise et al. 1998), which is native from the region of the Indian Ocean and is not present in Africa (Lecomte 1924, Jones 2005). How many generations was the stock raised on this media? If more than one, is it possible that you have forced selection pressures on the fly lines to be octanoic acid resistant? Additionally, are they allowed to only feed on ripe fruit, or rotten as well?

2) The Methods state that 3 different lines of *D. sechellia* were acquired from the species stock center. However, I cannot see which lines were used for what figures. Could you please clarify how many lines were tested, and what lines were used for the experiments? Was only one line tested, or was an outbred population bred? Having multiple lines to make the result generalizable is important. At least one additional *D. melanogaster* line would be preferred as well in order to make the overall claims of the paper more generalizable.

3) The authors state several times that genes involved in biogenic amine dopamine (DA) is “differentially expressed when compared to generalist Drosophila siblings” followed by two citations. I found the need to look at the citations, as the language of the sentence confused me. It could be referring to *D. sechellia's* relationship to *D. melanogaster*, *D. simulans*, or *D. mauritiana* (or some combination thereof). Please make this clear.

4) “We screened for genes that differ in expression between *D. sechellia* and its generalist sister species, *D. simulans*. We also screened for genes that are differentially expressed in *D. sechellia* when these flies chose their preferred host vs. when they were forced onto other food. *D. sechellia* increases expression of genes involved with oogenesis and fatty acid metabolism when on its host.”

Two peers are cited. Please provide values and the comparisons (species wise) from the papers cited here.

5) The authors state that fresh and overripe Morinda contains l-DOPA in equal amounts. Please show this data in supplement to confirm to readers that the state of the fruit being used (ripe/overripe) is not skewing the results.

6) Figure 2 demonstrates that depleting levels of l-DOPA in Morinda by using COMT hinders egg production stimulation. To further convey this synthetic alteration of the Morinda being the same as standardized food, please provide >S8/<S8 ratios as in Figure 2.

7) Figure 3 demonstrates that eggs of *D. sechellia* are larger in size when compared to those of sibling species. Please state what siblings are being compared to. Additionally, the eggs of *D. sechellia* fed Morinda are 3-fold larger than wild-type *D. melanogaster*. It is unclear to me what media the *D. melanogaster* lines are being fed in this experiment; banana food or standardized media? Given that the food substrate (and many other factors) is so important for these experiments, please explicitly write exactly what media each species is on, temperature, humidity, number of flies per vial/bottle, etc.

8) Figure 3 shows a Western blot of total protein whole-fly extracts for TH-PLE, Catsup, and ∝-tubulin. To make the claims more generalizable, please provide one additional strain for both *D. melanogaster* and *D. sechellia*. While it may be beyond the scope of the paper, it would be interesting to provide the other sister species on the Western blot as well (*D. simulans*, *D. mauritiana*).

9) Consider performing an LD50 assay with this DGRP line as this could be very useful to show that Catsup deficiencies could confer resistance to an otherwise toxic fruit, demonstrating multiple roles for this gene.

10) Figure 5 demonstrates the accumulation of TH-PLE in *D. sechellia* and *D. melanogaster* oocytes using antibody staining. D-E shows Catsup expression in nurse cells of *D. melanogaster*. If possible, accompanying images of *D. sechellia* and DGRP-357 ovaries should accompany the panels in this figure to further confirm Catsup expression changes.

11) Figure 6 shows that DA contributes to the behavioral resistance to Morinda, specifically to octanoic and hexanoic acid. Increased DA results in increased *D. melanogaster resistance*. I am having difficulty understanding how the experiment was performed. The Methods section does not provide any specifics of the apparatus used in the assay. Is it possible that the *D. melanogaster* are simply further away from the food if there is DA present? This question assumes the assay was performed in a cylindrical tube. Was it performed on a Petri dish where distance of the fly from the food would not be an issue? Please clarify and comment.

12) In the Methods section, under Behavioural assay, the authors state they used a mix of 3:1 octanoic to hexanoic acid. *M. citrifolia* contains highly volatile compounds that confer toxicity (Legal 1994), with the two most abundant being octanoic acid (58%) and hexanoic acid (19.24%) (Farine 1996). Octanoic acid is the most toxic compound in the ripe fruit (Legal 1994), which D. sechellia has been able to overcome by the development of both larval and adult tolerance mechanisms (R'Kha 1991, Legal 1992, Amlou 1997, Amlou 1998, Jones 1998, Jones 2001, Jones 2004). Given these observations, the ratio provided was on par. Please cite these papers so that readers are aware of the proper use of concentrations.

13) Figure 3—figure supplement 1 shows *D. sechellia* fed Morinda diet compared to 11 *Drosophila* siblings. More than one *D. sechellia* line should be tested if compared to multiple siblings. Additionally, stating what species the siblings are would be beneficial to the figure. When I looked up the reference (Markow 2009) the 11 species being referred to are:

a) *D. ananassae* (14024-0371.13), *D. erecta* (14021- 0224.01), *D. melanogaster* (14021-0231.36), *D. mojavensis* (15081-1352.22), *D. persimilis* (14011-0111.49), *D. pseudoobscura* (14011-0121.94), D. sechellia (14021- 0248.25), *D. simulans* (14021-0251.195), *D. virilis* (15010-1051.87), *D. willistoni* (14030-0811.24), *D. yakuba* (14021-0261.01). All cultures were maintained at 24^°^ C on a 12-h ⁄ 12-h light-dark cycle.

Earlier, the text uses the term “sibling species” only in reference to *D. simulans*, which is a sister to *D. sechellia* and *D. mauritiana*. This figure now uses the term for flies across the genus Drosophila. Please change the language to reflect the change in what flies are being discussed and add some text about why comparing to other species outside of the melanogaster subgroup is important. Additionally, please indicate which of the grey bars *D. sechellia* is from the Markow study.

14) Figure 4—figure supplement 2: sensory bristles contain a histogram of sternopleural bristles of *D. melanogaster* wild-type and *D. sechellia*. Please add in DGRP-357 to the histogram to draw the parallel between the two lines.

15) In the Methods section, under “Ovariole staining and immunofluorescence,” please define ON as overnight as it has not yet been defined in text.

16) The Materials and methods states that these experiments utilized three strain of *D. sechellia*. Were the Catsup gene sequences in all three of these strains? If so, do all three carry catsup alleles with a 45-bp deletion and the various non-synonymous mutations? If that is the case, it would suggest that these changes may have been present in the ancestor of present-day *D. sechellia* as it was evolving specialization on morinda. It is also possible that *D. sechellia* switched to morinda first as a breeding site, which then relaxed selection on Catsup, allowing it to accumulate the large deletion on non-synonymous mutations. It might be possible to examine details of molecular variation in and around Catsup to infer the age of the deletion, as well as to detect evidence of a recent selective sweep at this gene. If the alternative scenario I suggest (relaxed selection) had occurred, one would not expect to find evidence of a selective sweep.

---

## [Author Response]

*1) Please provide data to show that a morinda diet does not affect egg-laying in Drosophila melanogaster. Particularly as previous work shows that contact with the ripe fruit kills any member of the D. melanogaster complex with the exception of D. sechellia, which is attracted by and prefers it (Legal and Plawecki 1995)*.

As pointed out by the reviewers, morinda fruit has toxic effects on *D. melanogaster,* what hindered us to maintain a permanent culture of *D. melanogaster* in the fruit. For the purpose of this manuscript, we fed *D. melanogaster* a morinda diet for as long as each experiment required, which was not detrimental for the flies_ this fact has been clarified in the revised Materials and Methods and the citation suggested has been added.

Figure 1 shows that morinda does not significantly affect the relative production of eggs in *D. melanogaster* wild type Berlin compared to a standard diet. We have included the actual data of egg-production in a standard diet versus a morinda diet for *D. melanogaster* wild type Berlin and for one additional *D. melanogaster* line (Canton-S) in a supplementary figure (Figure 1—figure supplement 1). Results show that morinda has no effect or slightly inhibits egg-production in *D. melanogaster* wild type Berlin and *D. melanogaster* Canton-S, respectively.

*2) Please provide DAPI visualization of the ovaries to either confirm or deny apoptosis as a possible culprit for the depressed oviposition in* D. sechellia *when on standard media. Alternatively, some other method of detecting apoptosis, like TUNEL or activated caspase staining could be used to determine whether an oogenesis checkpoint has been triggered. Subsequently, demonstrating that apoptosis is no longer present when the flies are raised on Morinda fruit is also necessary. This is required because stressors can cause egg chambers in the ovaries to be eliminated by apoptosis at oogenesis checkpoints in region-2/3 of the germarium or stage 7/8 egg chambers (the mid-oogenesis checkpoint) (Drummond-Barbosa 2001; McCall 2004), which could be a possible explanation for the deceased egg numbers.*

Following the reviewers suggestion, we have checked for apoptosis occurring in *D. sechellia* ovaries, staining live ovaries with the vital dye acridine orange, following the protocol described in Arama and Steller (2006) (reference has been included therein). We found massive apoptosis occurring at stage S7/S8 cysts (mid-oogenesis checkpoint) and to a lesser extent in region-2/3 of the germarium, in *D. sechellia* fed a standard diet; confirming apoptosis as a possible culprit for the depressed oviposition on standard media. On the other hand, *D. sechellia* flies fed a morinda diet showed very few apoptotic cysts, in line with the stimulatory effect of morinda in egg-production. Corresponding images of the stained ovaries have been included in a supplementary figure (Figure 1—figure supplement 2). Additionally, we show that supplementation of a standard diet with l-DOPA is sufficient to reduce apoptosis in *D. sechellia* ovaries, while depleting morinda of l-DOPA by pre-treatment of the fruit with COMT prevents the anti-apoptotic effect of the morinda diet. We have quantified these effects and included the data in the revised Figure 2. In sum, these results further confirm the requirement of l-DOPA present in morinda fruit for an optimal reproduction of *D. sechellia*.

*3) Please test whether heteroallelic* D. melanogaster *fly lines carrying Catsup^ln270del^ and the null Catsup^1^ allele, show egg depression that can be rescued by feeding the flies l-DOPA (as with the* D. sechellia *experiments). This is a very important experiment that would complete the story.*

Figure 4 shows that heteroallelic *D. melanogaster* fly lines carrying *Catsup^In270Del^* and the null *Catsup^1^* allele show egg-depression compared to *D. melanogaster* wild type Berlin and to the parental lines. We have added new data showing that supplementation of a standard diet with l-DOPA significantly increases egg-production in *D. melanogaster* heteroallelic *Catsup^In270Del^/Catsup^1^* flies (Figure 4—figure supplement 4), as it does for *D. sechellia* (Figure 2).

*4) The paper states that no* D. melanogaster *wild-type eggs hatched in Morinda pulp but 13.5% of DGRP-357 eggs do. Could this be a strain difference in* D. melanogaster *lines? This issue could be addressed by testing a few other* D. melanogaster *lines or by genetic rescue.*

To rule out strain differences in *D. melanogaster* eggs on the ability to hatch in morinda fruit, we tested 3 further *D. melanogaster* wild type stains (Canton-S, Oregon-R and a wild line *D. melanogaster* pferta, caught in a local winery); one additional *D. melanogaster* line carrying the *Catsup^In270Del^* allele (DGRP-437) and two *D. sechellia* sister species *D. simulans* and *D. mauritiana*. The data obtained confirms that eggs carrying *Catsup^In270Del^* allele are able to hatch to living larvae 1 and even molt up to larvae 3 in morinda, while no *D. melanogaster* wild type egg made it alive to larvae 1; although some embryos completed development and hatched, dying in their way out of the chorion. Results have been graphed in a supplementary figure (Figure 5—figure supplement 1).

*5) At the end of the Results and Discussion please present a clear and complete description of the evolutionary scenario by which* D. sechellia *first became tolerant to and then dependent on morinda fruit. Even if the scenario includes some speculations, it is important to see the whole thing put together.*

The text has been changed accordingly. “Our results are compatible with an evolutionary scenario in which an original *Catsup* allele carrying a 45-bp deletion and six non-synonymous mutations was present in the ancestor of present-day *D. sechellia* (Figure 4—figure supplement 5), probably showing differentially low expression of CATSUP compared to *D. melanogaster* and the sister species *D. simulans* and *D. mauritiana* (see Figure 4 and Figure 4—figure supplement 1); what caused diminished egg production and concomitantly enhanced early survival in the fruit. A second event, further elevating DA (see Figure 4), would have redeemed adult *D. sechellia* from the toxic effects of morinda acids allowing them to oviposit in ripe fruits, benefiting from morinda stimulating effect on egg production and from the lack of siblings competition, rendering *D. sechellia* dependent on morinda. This dependency would have, in turn, shaped the super specialization of the chemosensory system (*6*, *7, 11*) present in today’s *D. sechellia* morinda obligate specialist.”

*Minor comments to which we request a constructive response*:

*1) The authors state that some* D. sechellia *in*
Figure 1
*were raised on a Morinda diet (pulp from the fruit collected from the greenhouse). On its native islands,* D. sechellia *specializes on the ripe fruit of Morinda citrifolia (Louis and David 1986, Lachaise et al. 1998), which is native from the region of the Indian Ocean and is not present in Africa (Lecomte 1924, Jones 2005). How many generations was the stock raised on this media? If more than one, is it possible that you have forced selection pressures on the fly lines to be octanoic acid resistant? Additionally, are they allowed to only feed on ripe fruit, or rotten as well?*

*D. sechellia* flies used in this work were originally obtained from the Drosophila Species Stock Center (UC, San Diego) and kept as a stock in standard *Drosophila* media. A subculture of this was used to establish a stock in morinda fruit; only ripe fruits were used; which was raised for more than 10 generations. To rule out a forced selection pressure on *D. sechellia* to be resistant to octanoic acid, we compared their survival in morinda fruit to that of *D. sechellia* flies from a new re-ordered stock of the same strain line (14021-0248.25). Results show no difference in survival to morinda between the new and old experimental stock, discarding an artificial adaptation to morinda acids in our experimental *D. sechellia* stocks. This fact has been clarified in the revised Material and methods.

*2) The Methods state that 3 different lines of* D. sechellia *were acquired from the species stock center. However, I cannot see which lines were used for what figures. Could you please clarify how many lines were tested, and what lines were used for the experiments? Was only one line tested, or was an outbred population bred? Having multiple lines to make the result generalizable is important. At least one additional* D. melanogaster *line would be preferred as well in order to make the overall claims of the paper more generalizable.*

We have tested one additional *D. sechellia* line and one additional *D. melanogaster* line for the main experiments, and have clearly stated in each revised figure legend which line of *D. sechellia* or *D. melanogaster* was used in each case.

*3) The authors state several times that genes involved in biogenic amine dopamine (DA) is “differentially expressed when compared to generalist* Drosophila *siblings” followed by two citations. I found the need to look at the citations, as the language of the sentence confused me. It could be referring to* D. sechellia's *relationship to* D. melanogaster*,* D. simulans*, or* D. mauritiana *(or some combination thereof). Please make this clear.*

The text has been clarified accordingly “Moreover, genes involved in DA differentiation have been shown differentially expressed in *D. sechellia* compared to the generalists species *D. melanogaster* and the sister species *Drosophila simulans* (Dworkin and Jones, 2009, Wurmser et al., 2011).”

*4) “We screened for genes that differ in expression between* D. sechellia *and its generalist sister species,* D. simulans*. We also screened for genes that are differentially expressed in* D. sechellia *when these flies chose their preferred host vs. when they were forced onto other food*. D. sechellia *increases expression of genes involved with oogenesis and fatty acid metabolism when on its host.”*

*Two peers are cited. Please provide values and the comparisons (species wise) from the papers cited here*.

The findings of Dworkin and Jones 2009 have been described in the revised Introduction and cited therein: “In turn, morinda stimulates egg production, and *D. sechellia* clearly prefers to oviposit in medium containing morinda carboxylic acids. On its host, *D. sechellia* increases expression of genes involved with oogenesis and fatty acid metabolism (Dworkin and Jones 2009). Thus, we here examined the dependence of *Drosophila sechellia* on morinda, for optimal reproduction.”

The findings of Wurmser et al 2011 have been described in the revised Results and cited therein: “In a screen for genes that differ in expression between *D. sechellia* and four geographically distinct populations of its generalist sister species *D. simulans*, Wurmser et al showed lower expression of the DA catabolic enzyme (*Dopamine-N-acetyltransferase*)…”

*5) The authors state that fresh and overripe Morinda contains l-DOPA in equal amounts. Please show this data in supplement to confirm to readers that the state of the fruit being used (ripe/overripe) is not skewing the results*.

The data of unripe and overripe fruits has been included in the text, showing similar values.

*6)*
Figure 2
*demonstrates that depleting levels of ^l^-DOPA in Morinda by using COMT hinders egg production stimulation. To further convey this synthetic alteration of the Morinda being the same as standardized food, please provide >S8/<S8 ratios as in*
Figure 2.

We have repeated the experiment feeding *D. sechellia* morinda or morinda pre-treaded to COMT and quantified *>S8/<S8*, confirming that depleting levels of l-DOPA in morinda by pre-incubation with COMT hinders stimulation of oogenesis, as quantified by >S8/<S8. The data has been included as a graph in the revised Figure 2—figure supplement 1.

*7)*
Figure 3
*demonstrates that eggs of* D. sechellia *are larger in size when compared to those of sibling species. Please state what siblings are being compared to. Additionally, the eggs of* D. sechellia *fed Morinda are 3-fold larger than wild-type* D. melanogaster*. It is unclear to me what media the* D. melanogaster *lines are being fed in this experiment; banana food or standardized media? Given that the food substrate (and many other factors) is so important for these experiments, please explicitly write exactly what media each species is on, temperature, humidity, number of flies per vial/bottle, etc.*

The text stating which siblings of *D. sechellia* are being compared to has been changed accordingly in the revised results: “Eggs of *D. sechellia* are characteristically large compared to those of sibling species *D. melanogaster*, *D. simulans*, *Drosophila ananassae*, *Drosophila erecta*, *Drosophila mojavensis*, *Drosophila persimilis Drosophila pseudoobscura*, *Drosophila virilis*, *Drosophila willistoni* and *Drosophila yakuba* (Figure 3 and Figure 3—figure supplement 1)”.

The feeding media of *D. melanogaster* to which *D. sechellia* egg size is being compared to has been explicitly written in the revised Results “In accordance, we observed *D. sechellia* eggs to be 45% larger in size compared to eggs of *D. melanogaster* wild type Berlin (Figure 3), in a standard diet condition. Upon being fed a morinda diet, eggs of *D. sechellia* increased in volume two-fold compared to eggs of conspecifics fed standard medium (Figure 3 and Figure 3—figure supplement 1); the resulting eggs had an almost three-fold larger volume than did *D. melanogaster* eggs in standard conditions (Figure 3)”.

Breeding conditions (food substrate, temperature, humidity, number of flies per vial/bottle, etc.) have been clearly stated in the revised Materials and methods and figure legends.

*8)*
Figure 3
*shows a Western blot of total protein whole-fly extracts for TH-PLE, Catsup, and* ∝*-tubulin. To make the claims more generalizable, please provide one additional strain for both* D. melanogaster *and* D. sechellia*. While it may be beyond the scope of the paper, it would be interesting to provide the other sister species on the Western blot as well (*D. simulans*,* D. mauritiana*).*

We have performed a Western blot of total protein whole-fly extracts for TH-PLE, CATSUP, and ∝-TUBULIN in one additional strain of *D. melanogaster*, one additional strain of *D. sechellia*, one strain of each of the sister species *D. simulans* and *D. mauritiana*, and one *D. melanogaster* stain carrying *Catsup^In270Del^* allele. The Western blot image and the relative quantification of proteins have been included in a supplementary figure (Figure 4—figure supplement 1).

*9) Consider performing an LD50 assay with this DGRP line as this could be very useful to show that Catsup deficiencies could confer resistance to an otherwise toxic fruit, demonstrating multiple roles for this gene*.

A survival analysis of DRGP-357 flies in morinda fruit showed no statistically difference with that of *D. melanogaster*, suggesting that further genes might be involved in the adult resistance of *D. sechellia* to morinda.

*10)*
Figure 5
*demonstrates the accumulation of TH-PLE in* D. sechellia *and* D. melanogaster *oocytes using antibody staining. D-E shows Catsup expression in nurse cells of* D. melanogaster*. If possible, accompanying images of* D. sechellia *and DGRP-357 ovaries should accompany the panels in this figure to further confirm Catsup expression changes.*

The corresponding images of *D. sechellia* and DGRP-357 ovaries showing a reduced CATSUP expression in the nurse cells and in the secretory cells of the spermatheca have been added to the revised Figure 5.

*11)*
Figure 6
*shows that DA contributes to the behavioral resistance to Morinda, specifically to octanoic and hexanoic acid. Increased DA results in increased* D. melanogaster *resistance. I am having difficulty understanding how the experiment was performed. The Methods section does not provide any specifics of the apparatus used in the assay. Is it possible that the* D. melanogaster *are simply further away from the food if there is DA present? This question assumes the assay was performed in a cylindrical tube. Was it performed on a Petri dish where distance of the fly from the food would not be an issue? Please clarify and comment.*

The chamber used is sufficiently small (35 mm diameter x 50 mm high) to assure a uniform exposure to the acids. Additionally, we have supplied the flies with a drop of fresh yeast paste to force the attracted flies to get in contact to the agar containing the mixture of morinda acids. Details on the methodology utilized have been clarified in the revised Materials and methods section.

*12) In the Methods section, under Behavioural assay, the authors state they used a mix of 3:1 octanoic to hexanoic acid.* M. citrifolia *contains highly volatile compounds that confer toxicity (Legal 1994), with the two most abundant being octanoic acid (58%) and hexanoic acid (19.24%) (Farine 1996). Octanoic acid is the most toxic compound in the ripe fruit (Legal 1994), which* D. sechellia *has been able to overcome by the development of both larval and adult tolerance mechanisms (R'Kha 1991, Legal 1992, Amlou 1997, Amlou 1998, Jones 1998, Jones 2001, Jones 2004). Given these observations, the ratio provided was on par. Please cite these papers so that readers are aware of the proper use of concentrations.*

The facts and papers mentioned have been cited in the revised Materials and methods.

*13)*
Figure 3—figure supplement 1
*shows* D. sechellia *fed Morinda diet compared to 11* Drosophila *siblings. More than one* D. sechellia *line should be tested if compared to multiple siblings. Additionally, stating what species the siblings are would be beneficial to the figure. When I looked up the reference (Markow 2009) the 11 species being referred to are: a)* D. ananassae *(14024-0371.13),* D. erecta *(14021- 0224.01),* D. melanogaster *(14021-0231.36),* D. mojavensis *(15081-1352.22),* D. persimilis *(14011-0111.49),* D. pseudoobscura *(14011-0121.94),* D. sechellia *(14021- 0248.25),* D. simulans *(14021-0251.195),* D. virilis *(15010-1051.87),* D. willistoni *(14030-0811.24),* D. yakuba *(14021-0261.01). All cultures were maintained at 24*^*°*^
*C on a 12-h ⁄ 12-h light-dark cycle.*

*Earlier, the text uses the term “sibling species” only in reference to* D. simulans*, which is a sister to* D. sechellia *and* D. mauritiana*. This figure now uses the term for flies across the genus Drosophila. Please change the language to reflect the change in what flies are being discussed and add some text about why comparing to other species outside of the melanogaster subgroup is important. Additionally, please indicate which of the grey bars* D. sechellia *is from the Markow study.*

Data from one additional *D. sechellia* line as well as the names of the 11 Drosophila siblings showed in the figure have been explicitly mentioned in the revised Figure 3—figure supplement 2 and its legend. The *D. sechellia* line from the Markow et al. study has been indicated. The language has been changed accordingly in the revised Results to refer to sister species (*D. simulans* and *D. mauritiana*) or “sibling species” (the Drosophila species here named).

*14)*
Figure 4—figure supplement 2*: sensory bristles contain a histogram of sternopleural bristles of* D. melanogaster *wild-type and* D. sechellia*. Please add in DGRP-357 to the histogram to draw the parallel between the two lines.*

We have added two additional *D. sechellia* lines, one additional *D. melanogaster* wild type line and two *D. melanogaster* lines carrying *Catsup^In270Del^* allele to the histogram of sternopleural bristles shown in the revised Figure 4—figure supplement 3. A graph showing the mode of sensory bristles for each species has been added to the same revised figure to aid comparison between species.

*15) In the Methods section, under “Ovariole staining and immunofluorescence,” please define ON as overnight as it has not yet been defined in text*.

ON has been defined as overnight in the revised Materials and methods.

*16) The Materials and methods states that these experiments utilized three strain of* D. sechellia*. Were the Catsup gene sequences in all three of these strains? If so, do all three carry catsup alleles with a 45-bp deletion and the various non-synonymous mutations? If that is the case, it would suggest that these changes may have been present in the ancestor of present-day* D. sechellia *as it was evolving specialization on morinda. It is also possible that* D. sechellia *switched to morinda first as a breeding site, which then relaxed selection on Catsup, allowing it to accumulate the large deletion on non-synonymous mutations. It might be possible to examine details of molecular variation in and around Catsup to infer the age of the deletion, as well as to detect evidence of a recent selective sweep at this gene. If the alternative scenario I suggest (relaxed selection) had occurred, one would not expect to find evidence of a selective sweep.*

We cloned *Catsup* from five *D. sechellia* strains original from two different original geographical locations. All five strains carry *Catsup* alleles with a 45-bp deletion and five conserved non-synonymous mutations (out of a total of nine); suggesting that these changes may have been present in the ancestor of present-day *D. sechellia* as it was evolving specialization on morinda. This result is discussed in the revised Discussion.